

# Solar Backscatter Ultraviolet (BUV) Retrievals of Mid-Stratospheric Aerosols from the 2022 Hunga Eruption

Robert James Duncan Spurr[1], Matt Christi[2], Nickolay Anatoly Krotkov[3], Won-Ei Choi[4], Simon Carn[5], Can Li[3], Natalya Kramarova[3], David Haffner[6], Eun-Su Yang[7], Nick Gorkavyi[7], Alexander Vasilkov[7], Krzysztof Wargan[7,3], Omar Torres[3], Diego Loyola[8], Serena Di Pede[9], Joris Pepijn Veefkind[9,10], and Pawan Kumar Bhartia[3*]

[1]RT Solutions, Inc., Cambridge, MA 02138, USA
[2]Independent Researcher, Fort Collins, CO 80524, USA
[3]NASA Goddard Space Flight Center, Greenbelt, MD 20771, USA
[4]NASA Postdoctoral Program, Oak Ridge Associated Universities, Oak Ridge, TN 37830, USA
[5] Michigan Technological University, Houghton, MI 49931, USA
[6] Independent Researcher, Glen Ellen, CA 95442, USA
[7] Science Systems and Applications, Inc., Lanham, MD 20706, USA
[8] German Aerospace Centre (DLR), Oberpfaffenhofen, 82234 Wessling, Germany
[9]Royal Netherlands Meteorological Institute (KNMI), 3730 AE De Bilt, The Netherlands
[10]Delft University of Technology, 2628 CN Delft, The Netherlands

*Emeritus

*Correspondence to*: Nickolay A. Krotkov (Nickolay.a.krotkov@nasa.gov)

**Abstract.** On January 15, 2022, a highly explosive eruption of the submarine Hunga volcano (Kingdom of Tonga) generated the largest stratospheric hydration event ever observed and the largest aerosol perturbation since the 1991 Pinatubo eruption. Here, we develop a novel method for satellite retrieval of stratospheric aerosol optical depth (AOD) and layer peak height ($Z_p$) using solar backscattered ultraviolet (BUV) radiation; this is made possible by the exceptional mid-stratospheric altitude of the Hunga aerosols. We analyze BUV observations of the Hunga stratospheric aerosol cloud on January 17, 2022 (47 hours after the eruption), using UV band 1 measurements from the TROPOspheric Monitoring Instrument (TROPOMI) on board the ESA/Copernicus Sentinel-5 precursor (S5P) satellite and the Ozone Mapping and Profiling Suite- Nadir Profiler (OMPS-NP) on board the NOAA-20 satellite. We retrieve AOD and $Z_p$ by fitting hyperspectral BUV radiance ratios in a narrow spectral window restricted to 289–296 nm, chosen in order to reduce interference from tropospheric clouds while highly sensitive to stratospheric aerosols located above ozone maximum altitude. The retrieval employs radiative transfer calculations from the Vector Linearized Discrete Ordinate Radiative Transfer (VLIDORT) forward model. We assume a single Hunga aerosol layer composed of polydisperse sulfuric acid spherical particles embedded in a Rayleigh atmosphere with a known ozone profile. The ozone profile is supplied from a version of the MERRA-2 Stratospheric Composition Reanalysis of the Microwave Limb Sounder (MLS) on board NASA EOS Aura satellite — produced by NASA's Global Modeling and Assimilation Office using a stratospheric chemistry model and MERRA-2 meteorology. We also include a dynamic $SO_2$ layer, which coincides spatially





with the retrieved aerosol vertical profile, and with the total loading normalized to the stratospheric SO$_2$ vertical column density from the operational TROPOMI SO$_2$ product. We validate our AOD retrievals against ground-based AERONET direct-sun AOD measurements as well as co-located OMPS-NP retrievals, and $Z_p$ retrievals against Cloud-Aerosol Lidar with Orthogonal Polarization (CALIOP) overpasses using Lagrangian trajectory modeling. We estimate the total Hunga stratospheric "wet" aerosol mass to be $M_{aer} \sim 0.5 \pm 0.05$ Tg. This value is consistent with our previous BUV estimates of Hunga gaseous sulfur dioxide (SO$_2$) emissions ($\sim 0.5$ Tg SO$_2$), and with the rapid conversion of SO$_2$ to sulfuric acid (sulfate) aerosol during the initial plume dispersion (SO$_2$ e-folding time $\sim$ 6 days), and $\sim$0.5 acid mass fraction in aqueous sulfuric acid solution.

## 1 Introduction

A paroxysmal eruption of the submarine Hunga volcano (Kingdom of Tonga; 20.550°S, 175.385°W) at ~04:15UTC on January 15, 2022 produced a steam-driven eruption column up to ~58 km altitude (Carr et al., 2022; Millán *et al*., 2022) and injected a massive plume of water vapor, sulfur dioxide (SO$_2$), and aerosols directly into the Southern tropical stratosphere (Carn et al., 2022; Khaykin et al., 2022; Millán et al., 2022; Legras et al., 2022; Schoeberl et al., 2022; Sellitto et al., 2022; Taha et al., 2022; Vömel et al., 2022). This was the largest volcanic explosion since Pinatubo in 1991, with a designated Volcanic Explosivity Index (VEI) of 5–6. The Hunga eruption produced enormous umbrella cloud(s) with diameter(s) >500 km, global Lamb waves (Kubota et al., 2022; Matoza et al., 2022), regional volcanic ash fall (Kelly et al., 2024), and Pacific-wide tsunamis (Lynett et al., 2022; Shrivastava et al., 2023).

Although volcanic ash ejecta remained at relatively low altitudes and quickly fell out over the Tonga area (Kelly et al., 2024), sub-micron non-absorbing, non-depolarizing sulfate-type aerosol particles persisted in the mid-stratosphere (Khaykin et al., 2022; Sellitto et al., 2022; Taha et al., 2022; Baron et al., 2023; Bernath et al., 2023; Bian et al, 2023; Boichu et al., 2023; Bourassa et al., 2023; Duchamp et al., 2023; Manney et al., 2023; Kahn et al., 2024; Stocker et al., 2024; Sichard et al., 2025). Discussions on the H$_2$O-accelerated conversion of volcanic SO$_2$ to sulfate aerosol may be found in previous studies (Carn et al., 2022; Legras et al., 2022; Sellitto et al., 2022; Zhu et al., 2022; Asher et al., 2023, Boichu et al., 2023; Sadehi et al., 2025; Stenchikov et al., 2025).

Initial estimates of the Hunga SO$_2$ emissions from solar backscatter ultraviolet (BUV) near-nadir satellite measurements did not account for interference from stratospheric aerosols; indeed, unexpectedly low amounts were reported for an eruption of this magnitude: ~0.5 Tg SO$_2$ (Carn et al., 2022). Infrared (IR) satellite measurements from the Cross-track Infrared Sounder (CrIS) instrument (Hyman and Pavolonis, 2020) measured a similar amount: ~0.4 Tg SO$_2$ (Sadeghi et al., 2025); however, retrievals from the Infrared Atmospheric Sounding Interferometer (IASI) reported roughly double this amount, i.e., >1 Tg SO$_2$ (Sellitto et al., 2022, 2024; Bruckert et al., 2025). These conflicting estimates provide a strong motivation to re-analyze BUV SO$_2$ measurements with explicit consideration of Hunga aerosol interference. To do this, we need first to introduce a suitable



Hunga aerosol optical model in the UV and then develop new quantitative BUV retrievals of non-absorbing stratospheric aerosol particles. Such nadir-BUV aerosol retrievals were not thought to be possible before the Hunga event; indeed, this is the first eruption in the modern satellite era to inject particles directly into the mid-stratosphere above the tropical ozone ($O_3$) density peak at ~25 km (Carr et al., 2022; Taha et al., 2022). This unique geophysical event provides access to solar backscattered shortwave UV radiation (< 300 nm wavelength) that is absorbed by ozone before reaching the lower altitudes

typical of volcanic aerosol injections. Another motivation for the present study is that during the early dispersion phase, the presence of Hunga aerosols significantly affected the BUV satellite retrievals of stratospheric ozone (Bhartia et al., 1993; Torres and Bhartia, 1995; Kramarova et al., 2024), as well as ocean color retrievals (Franz et al., 2024).

Figure 1 shows a Visible Infrared Imaging Radiometer Suite (VIIRS) true color map from January 17, 2022, two days after

the main Hunga eruption. The approximate locations of two distinct Hunga plumes are outlined over the Queensland region of Australia (Aerosol-rich plume) and over the Coral Sea ($SO_2$–rich plume) – see Fig. 5d in Carn et al. (2022). Also shown in Fig. 1 is a measurement track from the National Oceanic and Atmospheric Administration (NOAA)-20 Ozone Mapping and Profiler Suite (OMPS) Nadir Profiler (NP), which has a spatial resolution of 50 km at nadir. OMPS-NP is designed to measure stratospheric ozone profiles using BUV wavelengths below 300 nm, but as a nadir-sounding sensor it does not provide adequate

horizontal coverage of the Hunga aerosol plume. Nevertheless, the well-placed OMPS-NP overpass of the Hunga plume on January 17 (Fig. 1) did provide the first spectral observations of significantly enhanced BUV radiances produced by the Hunga aerosols at wavelengths below 300 nm.





**Figure 1: (Upper left)** Shortwave BUV radiances (270–310 nm) of aerosol-rich and background (aerosol-free) regions. **(Upper right)** Spectral radiance ratios (aerosol/background). **(Bottom)** True-color NOAA-20 VIIRS map of Australia and the Coral Sea on January 17, 2022. The solid line with colored segments shows the suborbital track of the NOAA-20 OMPS-NP. The inset panel at bottom left shows the variation of spectral radiance ratio with latitude measured by OMPS-NP in the Hunga aerosol plume. Dashed lines are CALIOP ground tracks—blue for daytime and black for nighttime. The yellow star indicates the approximate location of the Aerosol Robotic Network (AERONET) Lucinda site.

Accordingly, to estimate total Hunga aerosol optical depth (AOD), layer peak height, and column mass, we have analyzed

BUV observations taken by an imaging spectrometer — TROPOspheric Monitoring Instrument (TROPOMI) on board the

Copernicus Sentinel-5 precursor (S5P) satellite. TROPOMI is eminently suitable for this task, because of its contiguous daily

coverage and high spatial resolution (nominally 5.5 km by 28 km below 300 nm in band 1) (Ludewig et al., 2020). On January

17 at ~03:16 UTC (47 hours after the January 15 eruption) the bulk of the Hunga aerosol was observed over northeast Australia

(Fig. 1), while the bulk of the attendant $SO_2$ plume was trailing over the Coral Sea (see also Fig. 5d of Carn et al. (2022)).



In this paper we use such BUV spectral enhancements to develop a new BUV algorithm for non-absorbing AOD and peak-height retrievals and estimates of column aerosol mass. The retrieval algorithm is based on spectral fitting of hyperspectral BUV radiance ratios with the forward model driven by the Vector Linearized Discrete Ordinate Radiative Transfer

(VLIDORT) model (Spurr and Christi 2019) and combined with polydisperse Mie calculations of aerosol optical properties. Based on collocated Cloud-Aerosol Lidar with Orthogonal Polarization (CALIOP) backscatter measurements (Fig. 1), we have assumed a single aerosol layer composed of polydisperse homogeneous sulfuric acid spherical (non-depolarizing) particles embedded in a Rayleigh-scattering atmosphere with a known ozone profile for trace gas absorption. We also included a dynamic $SO_2$ vertical profile, assumed spatially coincident with the retrieved aerosol plume, but normalized to the TROPOMI

operational stratospheric $SO_2$ column density (Theys et al., 2017).

This paper begins by summarizing the TROPOMI UV band 1 UV measurements (Section 2). Section 3 describes our retrieval algorithm, comprising an overview of the retrieval strategy (Section 3.1), deployment of the VLIDORT-based forward model component (3.2), the inverse model (3.3), a discussion on aerosol optical properties and trace gas parameterization (3.4), and

a validation of the retrieval algorithm using synthetic data. In Section 4, we present our results on the retrieval and validation of aerosol peak height (Section 4.1) and AOD (4.2), followed by estimates of the aerosol column mass (4.3) and equivalent sulfur (S) mass (4.4). Section 4.5 contains comparisons with recent IR-based retrievals of $SO_2$ and sulfate aerosol mass from other studies of this unique event. Section 5 concludes with a summary of the paper along with final remarks.

## 2. Solar Backscatter Measurements in the UV (BUV)

This section is concerned with measurement data. First, we focus on the TROPOMI UV backscatter measurements used for the aerosol plume retrieval; this is followed by a discussion on anomalies in the satellite ozone record caused by the Hunga eruption.

### 2.1 TROPOMI Band-1 radiances and radiance ratios

A detailed description of the current version of the TROPOMI Level 1B radiance product (L1B_RA_BD1), including the data

file format in NETCDF-4 and the data fields, is given in the TROPOMI Level 1B product "readme" file (doi.org/10.5270/S5P-kb39wni, Ludewig et al., 2023). TROPOMI provides excellent spatial resolution and daily global coverage in the shortwave UV band 1 (267–300 nm); however, TROPOMI has known radiometric and sensor degradation issues in band 1 (Ludewig et al., 2020), and it has proved necessary to apply soft calibration techniques to improve the data quality. The TROPOMI soft calibration is computed from characterization of the differences between measured and modelled radiances (absolute

residuals), following a similar approach to that described in Mettig et al. (2021). Since this soft calibration is designed to correct input radiances for the TROPOMI ozone profile data product, forward Radiative Transfer (RT) model calculations





were performed over the combined spectral range of band 1 and 2 (270–330 nm). Pressure, temperature, and ozone profiles from the Copernicus Atmosphere Monitoring Service (CAMS) were used as inputs to the RT model. Additionally, CAMS ozone profiles were scaled to match total ozone columns derived from the independent OMPS Nadir Mapper gridded column

ozone data (Jaross 2017). The modelled atmosphere does not contain clouds or aerosols, but particulate scattering effects are compensated through adjustment of the surface albedo. With this in mind, we have fitted the scene albedo in a small spectral window (328–330nm) and assumed this albedo to be representative for the entire fitting window. Radiance residuals (measurement to model) are computed for a single seasonal TROPOMI orbit over the Pacific Ocean. To compute correction parameters, the radiance residuals are compiled separately for each year and then applied to the radiance measurements in that

year. Correction parameters are provided as a function of the TROPOMI cross-track position (i.e., CCD detector row), wavelength and radiance level; they are applied to the uncorrected radiance signal ($R_{uncorr}$) by subtracting the bias ($R_{corr} = R_{uncorr}$–bias). For TROPOMI orbits 22086 and 22087 affected by Hunga aerosols we apply the fixed correction available for the highest radiance.

One option for constructing the retrieval measurement vector is to use sun-normalized radiances, or equivalently the "N-

values" (defined as $N = -100\log_{10}\left(\frac{I}{F}\right)$ where $I$ denotes Earthshine radiance and $F$ solar irradiance), as shown in Fig. 2 for two affected TROPOMI orbits on January 17, 2022 at four band 1 wavelengths.

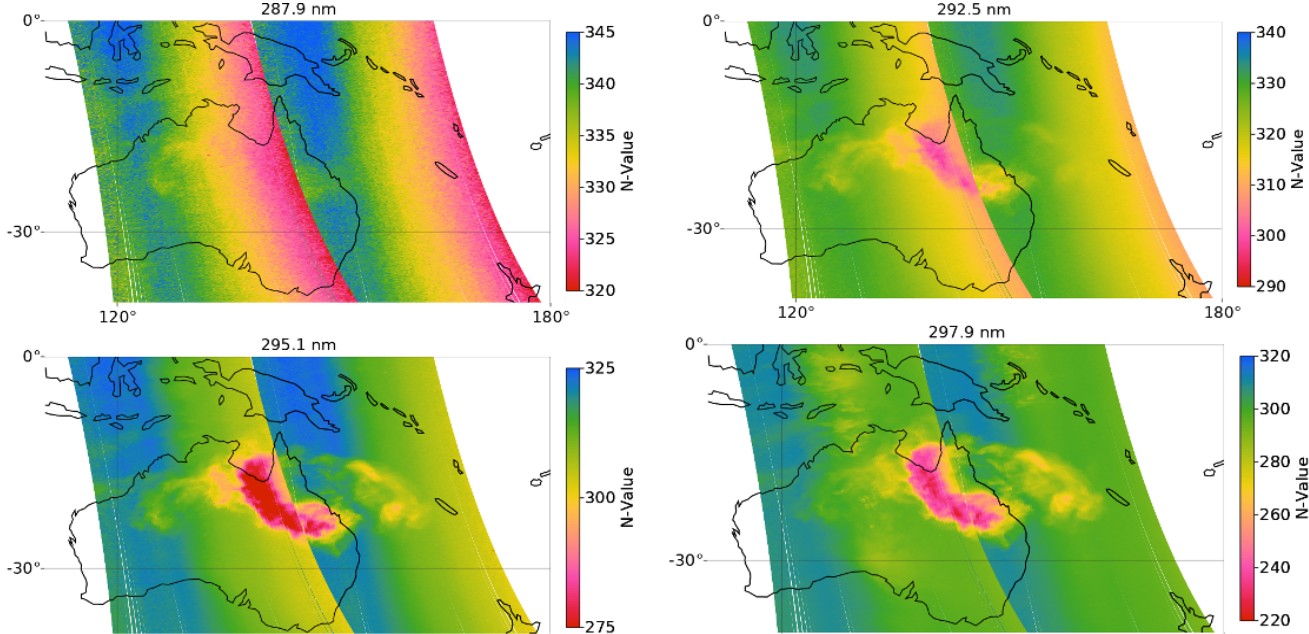

**Figure 2: Maps of the Hunga aerosol plume on January 17, 2022, using TROPOMI band 1 sun-normalized radiances before applying soft-calibration at four wavelengths below 300 nm (287.9, 292.5, 295.1 and 297.9 nm) in orbits 22086 (right-hand swath) and 22087 (left-hand swath). Plotted are the N-values, defined as $N = -100\log_{10}\left(\frac{I}{F}\right)$, where $I$ denotes Earthshine radiance, and $F$ the solar irradiance. Note the different N-value scales for different wavelengths.**





However, in order to emphasize the Hunga aerosol spectral signal and to reduce its dependence on TROPOMI band 1 calibration and degradation issues, we prefer to use BUV *radiance ratios*, in which the radiance values from the orbit 22086

and 22087 pixels are normalized to background radiances for the same cross-track pixels from an adjacent aerosol-free orbit, in this case TROPOMI orbit 22085. Radiance ratios are plotted in Fig.3. The advantage of normalizing to background radiances is that this pixel-wise division leads to a partial cancellation of known radiometric and degradation interferences from the TROPOMI band 1, as reported in Ludewig et al. (2020). We do account for the ozone profile differences between the orbits using MERRA-2 Stratospheric Composition Reanalysis of the Microwave Limb Sounder (MLS) on board NASA EOS Aura

satellite as described later in section 2.2. The normalization requires correction for tropospheric clouds if one uses full spectral window. Therefore, in this study we use short spectral fitting window from 289nm to 296nm which is not affected by bright tropospheric clouds (see example for background pixel 1 in Fig. 3).

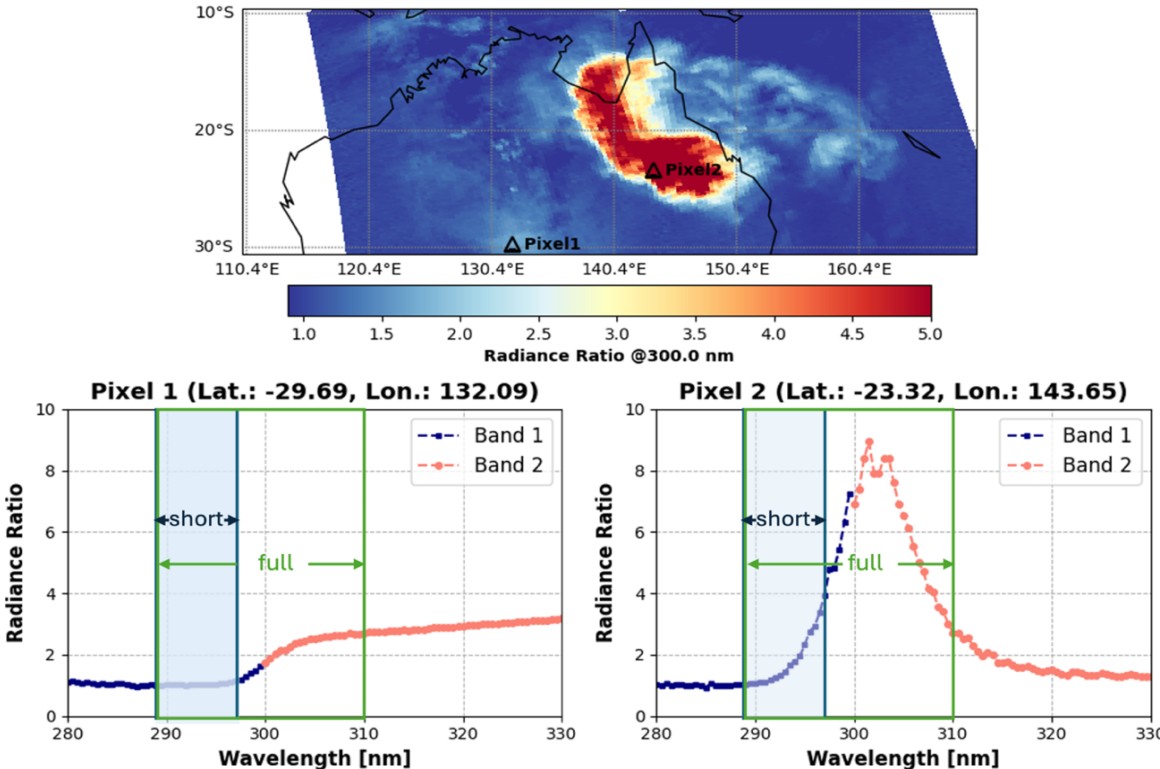

**Figure 3: (Upper panel) Map of normalized TROPOMI radiance at 300 nm on January 17, 2022. Normalized TROPOMI radiance spectra for a background (aerosol-free) pixel (Pixel 1; lower left) and a pixel in the Hunga aerosol plume (Pixel 2; lower right). The plots indicate the spectral coverage of TROPOMI UV bands 1 and 2 (with an overlap around 300 nm), as well as the 'short' and 'full' spectral fitting windows used for aerosol retrievals. Pixel 2 shows the typical enhancement of the radiance-ratio signal in the presence of Hunga aerosols, whereas Pixel 1 shows the effect of bright tropospheric clouds at longer wavelengths (>300 nm). Note that tropospheric clouds do not appear in the short window (Pixel 1; lower left). We note that bands 1 and 2 form two distinct halves of the detector. The (sharp) separation between them is halfway along the number of pixels in the spectral direction; this occurs at ~300 nm, but it also depends on the across-track location because of the spectral smile.**



For the 2-parameter (AOD and aerosol peak height, $Z_p$) aerosol retrievals in Section 4, we restrict the spectral fitting window
from 289 nm to 296 nm (hereafter, denoted as the 'short' window) to reduce interference from bright tropospheric clouds.
Figure 3 shows radiance ratios (using TROPOMI orbit 22085 as background) over a large part of Australia, with six- to ten-
fold enhancements of the radiance-ratio signal at 300nm in the presence of stratospheric aerosol. Background pixel ratios show
signals from tropospheric clouds, but ratios in the short wavelength window are free of such signals (Fig 3; this is also seen in
the lower right panel of Fig 2, where cloud signals at wavelength 297.9 nm are apparent in the Australian Bight near the south-
central coast).

Figure 4 presents spectral radiance ratio measurements at four different wavelengths. At wavelengths shorter than 288 nm
there is no Hunga aerosol signature because of strong absorption by ozone above the Hunga plume altitude. The major plume
signature over Northeast Australia becomes much clearer for the longer wavelengths, 296 to 298 nm, but even at 292 nm (top
right) there is evidence of a forward plume streamer at high altitude over northwest Australia. Also of interest are the
tropospheric cloud echoes near the south-central Australian coast, evident at the two longer wavelengths (lower plots), but not
present at shorter wavelengths. This indication has allowed us to set a cloud screening threshold at 296 nm. In Supplement S1,
we present an animated video of the full spectral scan of TROPOMI band 1 and band 2 radiance ratios from 280 to 330 nm in
steps of 0.5 nm (https://doi.org/10.5446/70186).


We contour the Hunga stratospheric aerosol plume using a Cloud Screening Index (CSI), which is defined as a radiance ratio
at 296 nm (see Appendix C for the CSI determination). Hereafter, we only consider TROPOMI pixels with CSI > 1.1 for
Hunga aerosol retrievals.

Note that we have also considered 3-parameter retrievals, with the third state vector element being the total column of $SO_2$
emitted simultaneously in the Hunga eruption; for this retrieval, we use the larger spectral fitting window (289–310 nm),
denoted as the 'full' window in the sequel. This retrieval takes advantage of strong $SO_2$ absorption features present in
TROPOMI band 2 radiances (306–310 nm) but it does require the application of a cloud-correction step. This retrieval is
currently under investigation.




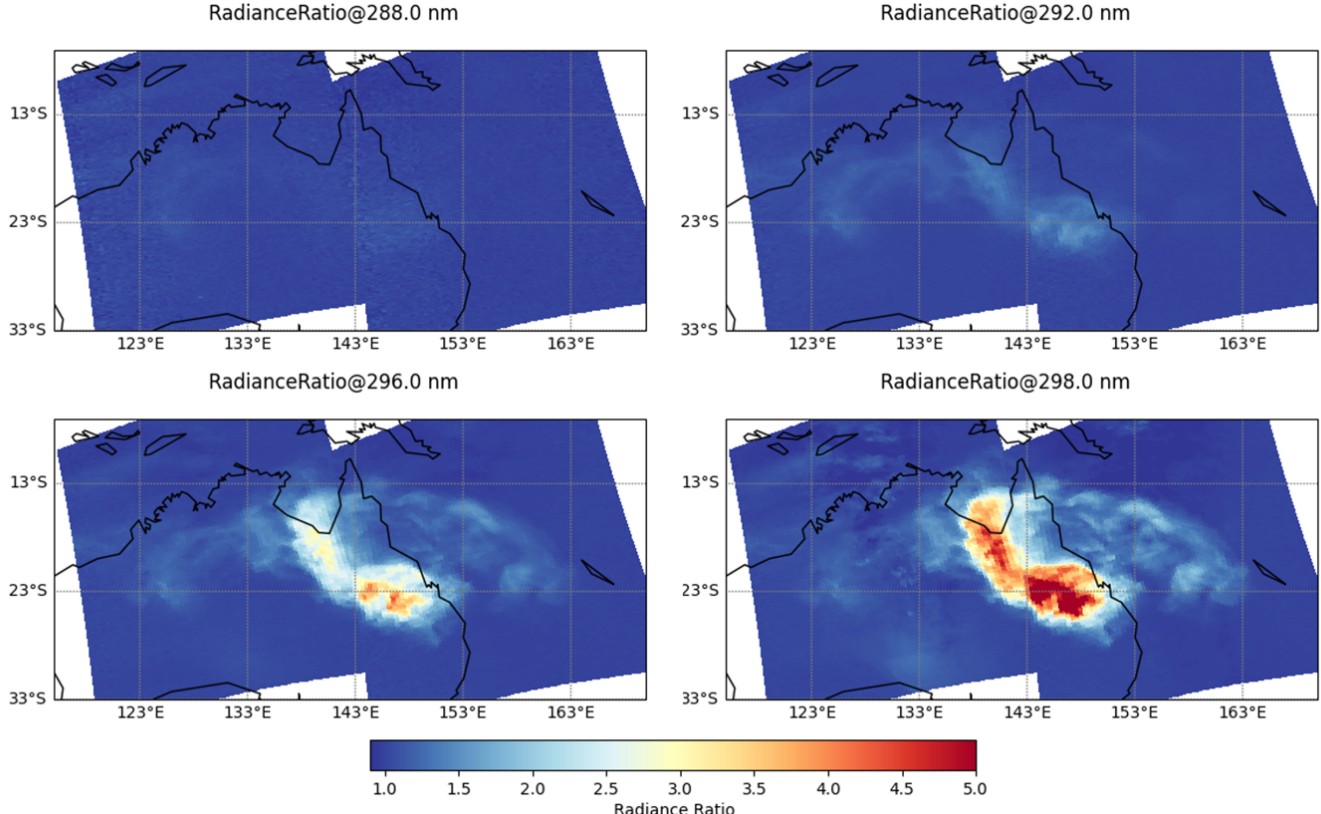

**Figure 4: Radiance ratios (orbits 22086/22085 and 22087/22085) at four wavelengths. See the full spectral movie in Video Supplement S1 (https://doi.org/10.5446/70186).**

## 2.2 Ozone Anomalies

As noted in the Introduction, we are motivated here to discuss the disruption in the stratospheric ozone record due to the Hunga aerosols. Anomalies in a number of total ozone column (TOC) products were observed in the presence of the Hunga eruption plume; notably, enhanced BUV scattering signals due to high-altitude aerosol from the Hunga plume were mistakenly interpreted as TOC depletion in BUV ozone retrievals from all instruments (OMI, OMPS, GOME-2, as well as TROPOMI). Mid-stratospheric aerosols are not represented in the forward models used for TOC retrievals, resulting in artificial TOC depletion during episodes of elevated volcanic aerosol loading. In this regard, it was necessary to re-analyze raw BUV spectra to flag pixels affected by the Hunga aerosols. Indeed, the presence of such 'bad-quality' ozone data will contaminate assimilation products that rely on BUV TOC data. For instance, artificially low BUV TOC data affected by Hunga plume resulted in anomalous assimilation outcomes in released M2-SCREAM reanalysis data (MERRA-2 Stratospheric Composition Reanalysis of Aura Microwave Limb Sounder (MLS) produced by NASA's Global Modelling and Assimilation Office using a stratospheric chemistry model and MERRA-2 meteorology: Gelaro et al., 2017; Wargan et al., 2023), as shown in the left panel of Fig. 5.




In contrast, ozone profile measurements from Microwave Limb Sounder (MLS) on board NASA Earth Observing System - chemistry (EOS Aura) satellite were not affected by the Hunga aerosols with effective radii ~0.2–0.4 $\mu m$ (Boichu et al., 2023; Duchamp et al., 2023). Although stratospheric ozone profiles are strongly constrained by MLS, anomalously low OMI total ozone affects the ozone profile to some degree in the assimilated system, because data assimilation distributes the analysis

increments in vertical levels according to a prescribed amount of background uncertainty. Also in this regard, there is some evidence (Evan et al. 2023; Zhu et al., 2023) of an actual physical ozone depletion inside the plume within days after the eruption.

For BUV aerosol plume retrievals, it is clear from the above discussion that we cannot use aerosol-contaminated BUV ozone

data. Instead, for the air density, temperature, and ozone profiles, we have used specially-processed M2-SCREAM reanalysis data. This special processing involves the removal of erroneous OMI-retrieved total ozone columns in the Hunga plume from this reanalysis (both OMI and MLS are on the Aura platform), for the period of 15–25 January 2022. Figure 5 illustrates the total ozone map obtained from this special reprocessing and compares it to the original assimilated product. Assimilated ozone columns show a marked anomaly over Northeast Australia when the assimilation system includes erroneous OMI TOC data

(Fig. 5, left). Filtering out the anomalous OMI TOC and performing the necessary reanalysis gives rise to a much smoother total ozone map (Fig. 5, center). Assimilated ozone profiles have also shown low anomalies in the stratosphere when erroneous OMI TOC values were included (Fig. 5, right).

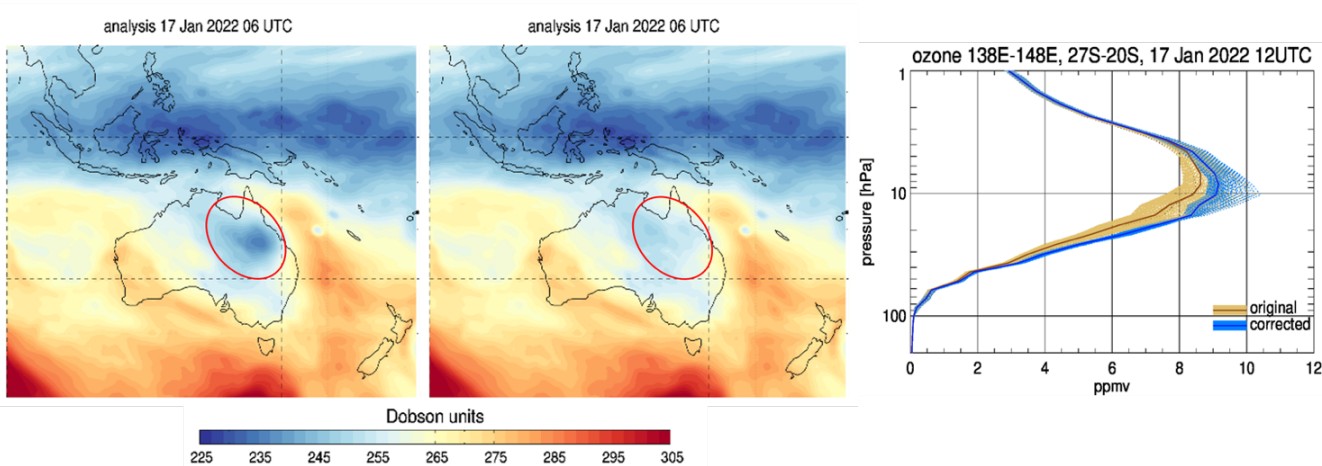

**Figure 5: Spatial distribution of TOC in M2-SCREAM reanalysis data on January 17, 2022 at 6 UTC. (Left) Assimilation using anomalous OMI V3 TOC retrievals. (Center) Assimilation after filtering out the anomalous OMI TOC re-processed M2-SCREAM. (Right) Ozone profiles from the original M2-SCREAM reanalysis data (orange), and from the assimilation that omitted anomalous OMI TOC (blue) on January 17, 2022 at 12 UTC.**



## 3. The Hunga Nadir-BUV Aerosol Retrieval Algorithm

In Section 3.1 we introduce measurement and state vectors, focusing on the use of ratioed BUV radiances from adjacent orbits; this section deals with parameterization of the Hunga plume in terms of a pseudo-Gaussian profile shape. We summarize the deployment of the VLIDORT Radiative Transfer (RT) model in the forward model component in Section 3.2, with special emphasis on the linearized optical property set-ups required by VLIDORT to generate analytically-derived Jacobians with respect to AOD and peak height. The least-squares inversion model is outlined in Section 3.3, and in Section 3.4, we discuss

the preparation of aerosol optical properties, parameterization of the $SO_2$ vertical profile, and the use of input ozone profiles from the modified MERRA2 reanalysis. Section 3.5 is concerned with algorithm validation using synthetic data.

### 3.1 Retrieval Algorithm: State and Measurement Vectors

Measurement vectors for the retrieval are ratios of two TROPOMI Earthshine spectra (Eq. 3.1):

$$M_{meas}(\lambda) = \frac{R_{meas}^{(Volc)}(\lambda)}{R_{meas}^{(Bkgd)}(\lambda)}. \qquad (3.1)$$

Here, the quantity $R_{meas}^{(Volc)}(\lambda)$ is a TROPOMI spectrum with a clearly-pronounced volcanic aerosol signal (e.g., from orbit 22086), and $R_{meas}^{(Bkgd)}(\lambda)$ is a similar spectrum from an adjacent background aerosol-free TROPOMI orbit (orbit 22085). As noted in Section 2.2, the key assumption is that the re-processed M2-SCREAM stratospheric ozone and temperature profiles are

accurately characterized regardless of the presence of Hunga aerosols. Geolocations for each spectrum in the orbit pair are very similar, and we assume the same wavelength grid for both orbits.

The dimension of the measurement vector is variable – this depends on the choice of spectral fitting window and the application of spectral smoothing (if any). For the short window (289–296 nm), there are typically 107 spectral elements without

smoothing.

For errors on the measurements, we take radiance noise levels from the TROPOMI Level 1b product, assuming individual measurement errors to be uncorrelated. The uncertainty in the radiance ratio is then calculated using error propagation, combining the relative uncertainties of each scene in quadrature and scaling the result by the radiance ratio.


The number of retrieved quantities determines the dimension of the state vector. The Hunga aerosol loading profile $\{E(z)\}$ as a function of altitude $z$ is taken to have a 'pseudo-Gaussian' shape (this analytic parameterization is sometimes called a Generalized Distribution Function), characterized by three parameters $\{A_0, z_p, h_w\}$. Here, $A_0$ is the Hunga stratospheric AOD at a fixed reference wavelength $\lambda_{ref}$ (taken to be 312 nm), $z_p$ is the plume peak height in [km], and $h_w$ is the half-width-half-



maximum of the plume profile (also in [km]). For our 2-parameter retrieval, the state vector is $\mathbf{x} = \{A_0, z_p\}$, with $h_w$ treated

as a known model parameter (see Section 3.4 for more on this quantity). Appendix A contains a description of this pseudo-

Gaussian parameterization, including explicit analytic expressions for $E(z; A_0, z_p, h_w)$ in terms of the three parameters of

interest, and a determination of analytic derivatives $\left\{\frac{\partial E(z)}{\partial A_0}, \frac{\partial E(z)}{\partial z_p}\right\}$ necessary for deriving Jacobian output from the

forward model component of the retrieval algorithm.


## 3.2 Forward Model Radiative Transfer (RT) and Analytic Jacobians

Next, we consider the forward-model RT simulation of the ratioed BUV spectra. The retrieval algorithm requires forward-

model Jacobians with respect to state vector elements; here, the 2-parameter vector $\mathbf{x} = \{A_0, z_p\}$. In general, the forward model

will generate simulations $M_{sim}(\lambda)$ to match the quantities in Eq. (3.1). This requires two RT simulations: $R_{sim}^{(Bkgd)}(\lambda)$ is the

RT calculation for a background atmospheric scenario with no aerosol, and $R_{sim}^{(Volc)}(\lambda)$ is RT simulation with similar

background but including the aerosol plume. In addition to $M_{sim}(\lambda)$, the forward model must also generate Jacobians with

respect to the aerosol parameters:

$$\mathbf{K}(\lambda) \equiv \left[\frac{\partial M_{sim}(\lambda)}{\partial A_0}, \frac{\partial M_{sim}(\lambda)}{\partial z_p}\right] = \frac{1}{R_{sim}^{(Bkgd)}(\lambda)} \cdot \left[\frac{\partial R_{sim}^{(Volc)}(\lambda)}{\partial A_0}, \frac{\partial R_{sim}^{(Volc)}(\lambda)}{\partial z_p}\right]. \quad (3.2)$$


We use the VLIDORT discrete-ordinate RT model for simulating polarized light fields (Spurr, 2006; Spurr and Christi, 2019)

for the two forward-model calculations required for these measurements of ratioed backscatter. $R_{sim}^{(Bkgd)}(\lambda)$ is a simulation for

a Rayleigh scattering atmosphere with $O_3$ absorption, and no aerosols or $SO_2$; the $O_3$ profile is from assimilation constrained

by MLS and corrected OMI measurements, as described in Section 2.2. The major advantage with VLIDORT is its

simultaneous ability to generate not only the radiance fields, but also any set of analytically-derived Jacobians (weighting

functions) with respect to atmospheric or surface parameters.

Aerosol optical properties are required for the $R_{sim}^{(Volc)}(\lambda)$ simulation based on a pseudo-Gaussian aerosol plume loading; for

this, we use a linearized Mie scattering model (Spurr et al., 2012) to develop these properties from assumed knowledge of

Hunga aerosol microphysical quantities (refractive index, particle size distribution parameters). Details of the aerosol optical

property Mie derivations are found below in Section 3.4, along with a discussion of other atmospheric constituent profiles (in

particular, $O_3$ and $SO_2$).





For radiance simulations, it is necessary to construct an input set of total optical properties (optical thickness values, single-scattering albedos, spherical-function expansion coefficients, scattering matrices) for VLIDORT. In addition, for calculations of associated Jacobians with respect to aerosol retrieval parameters, VLIDORT requires an additional set of *linearized* total optical property inputs. Determination of VLIDORT optical property inputs is discussed in Appendix B.

### 3.3 Inverse model

The retrieval inverse model is an iterative damped non-linear least-squares minimization using a modified version of the Levenberg-Marquardt (L-M) algorithm (LMA) (Marquardt, 1963) with variable step-size (see e.g., Chong and Zak, 2001). If $\mathbf{x}_m$ is the state vector at iteration $m$, then the estimate for the state vector at the next iteration is given by:

$$\mathbf{x}_{m+1} = \mathbf{x}_m + \alpha_m (\mathbf{K}^\mathrm{T} \boldsymbol{S}_\epsilon^{-1} \mathbf{K} + \mu_m \mathbf{I})^{-1} \mathbf{K}^\mathrm{T} \boldsymbol{S}_\epsilon^{-1} (\mathbf{y}_{meas} - \mathbf{F}(\mathbf{x}_m)) \qquad (3.3)$$

Here, $\mathbf{K}$ is the Jacobian matrix which has row vector $\mathbf{K}(\lambda_i)$ for wavelength $\lambda_i$ in the fitting window as seen in Eq. (3.2), $\mathbf{y}_{meas}$ is the measurement vector with entries $M_{meas}(\lambda_i)$ (Eq. (3.1)), $\mathbf{F}(\mathbf{x}_m)$ is the forward-model simulated measurement vector with entries $M_{sim}(\lambda_i)$ of the same form as that in Eq. (3.1), $\boldsymbol{S}_\epsilon$ is the measurement and forward-model error covariance matrix (here considered diagonal), and $\mathbf{I}$ is the identity matrix. The "T" superscript denotes matrix transpose.

The L-M damping parameter $\mu_m$ is adjusted as needed at each iteration in order to ensure the approximation to the Hessian matrix $(\mathbf{K}^\mathrm{T} \boldsymbol{S}_\epsilon^{-1} \mathbf{K} + \mu_m \mathbf{I})$ in Eq. (3.3) remains positive definite. This ensures that the shape of the cost-function approximation we are seeking to minimize during that iteration is "bowl-shaped", and that the negative of the gradient $\mathbf{K}^\mathrm{T} \boldsymbol{S}_\epsilon^{-1} (\mathbf{y}_{meas} - \mathbf{F}(\mathbf{x}_m))$ in Eq. (3.3) points in a direction to descend into the bowl.

The step-size $\alpha_m$ is sometimes determined by a line search, in order to *guarantee* the cost function approximation is minimized at each iterative step; however, this procedure would be too numerically expensive in our retrieval. Instead, in order to ensure that $A_0$ and $z_p$ remain in physical parameter space at each iteration step, we simply halve the step size repeatedly until this physicality condition is satisfied (starting at $\alpha_m = 1$)..

Convergence is reached when relative differences in state-vector elements between adjacent iterations are all below a threshold criterion ($10^{-2}$ in our case), and/or when the cost-function itself reaches a clear minimum in fitting space. Spectral points are $\{\lambda_i\}$, $i = 1, \ldots N_s$, where the number of points $N_s$ depends on the selection of TROPOMI measurements in UV Band 1. With two parameters ($A_0$ and $z_p$), matrix $\mathbf{K}$ in Eq. (3.3) has dimension $2 \times N_s$.



In addition to the above, to obtain *better* estimates of uncertainties on the retrieved state vector elements $A_0$ and $z_p$, a facility was implemented to modify the original standard deviations from the measurement and forward-model error covariance matrix $\mathbf{S}_\epsilon$ used in the retrieval (which is often based initially on measurement characteristics alone such as signal-to-noise ratio (SNR)). This was done as follows. As part of each original retrieval, a chi-square diagnostic $\chi^2 = [\mathbf{y}_{meas} - \mathbf{F}(\mathbf{x})]^\mathrm{T} \mathbf{S}_\epsilon^{-1} [\mathbf{y}_{meas} - \mathbf{F}(x)]$ is produced. If the value of chi-square for the retrieval is too large - indicating that the estimated

mismatch in actual measurements $\mathbf{y}_{meas}$ versus forward-model-simulated measurements $\mathbf{F}(\mathbf{x})$ is generally too large relative to that assumed using the original standard deviations in $\mathbf{S}_\epsilon$ - then the *expected value* of chi-square for the retrieval (i.e. the number of measurements $n$ used in the retrieval minus one), along with $\mathbf{y}_{meas}$ , the final values of $\mathbf{F}(\mathbf{x})$ from the original retrieval, and the original standard deviations used in $\mathbf{S}_\epsilon$, are used to compute an additional contribution to the estimated standard deviation of measurement/forward model error for each measurement. These contributions help to account for the

influence of unknown sources of measurement and/or forward model error. These values are then added to the original measurement standard deviations on the diagonal of $\mathbf{S}_\epsilon$ and the retrieval is then *re-run* using the more realistic combined standard deviations of measurement and forward-model error. With this procedure, retrieved values of the state vector elements often do not change significantly, but their estimated uncertainties are often more realistic (i.e. larger), along with improved chi-square diagnostics.


### 3.4 Aerosol optical properties, trace gas profiles and parameterizations, forward-model setups

*Background profiles*

As noted in Section 2.2, we use specially-processed ozone and temperature assimilated vertical profiles (reprocessed M2-SCREAM) – this data set contains pressure, temperature and ozone volume mixing ratios specified for a 72-layer vertical grid

at 1–2 km vertical resolution in the stratosphere. This 72-layer grid forms the basis for the atmospheric stratification. We have imposed a finer vertical resolution for the Hunga aerosol plume (typically 0.25 km is sufficient), in order to properly characterize the pseudo-Gaussian plume shape. Baseline retrievals are done assuming the aerosol peak height $z_p$ to be between 24 and 34 km, in general above the ozone density peak at ~25 km. We performed a sensitivity study allowing the minimum value of $z_p$ to be 20 km; we found that this had a negligible impact on the total aerosol mass retrieval. Figure 6 illustrates the

assumed aerosol and ozone profile distributions, along with CALIOP daytime overpasses from January 17, 2022. In particular, the CALIOP overpass at 5:29 UTC shows a narrow plume at 32 km (Fig. 6, right); based on this, we estimate a plume half-width of ~0.4 km, and this value was assumed throughout the retrievals discussed in the paper. The CALIOP data were used to validate our retrievals (see Section 4.4).





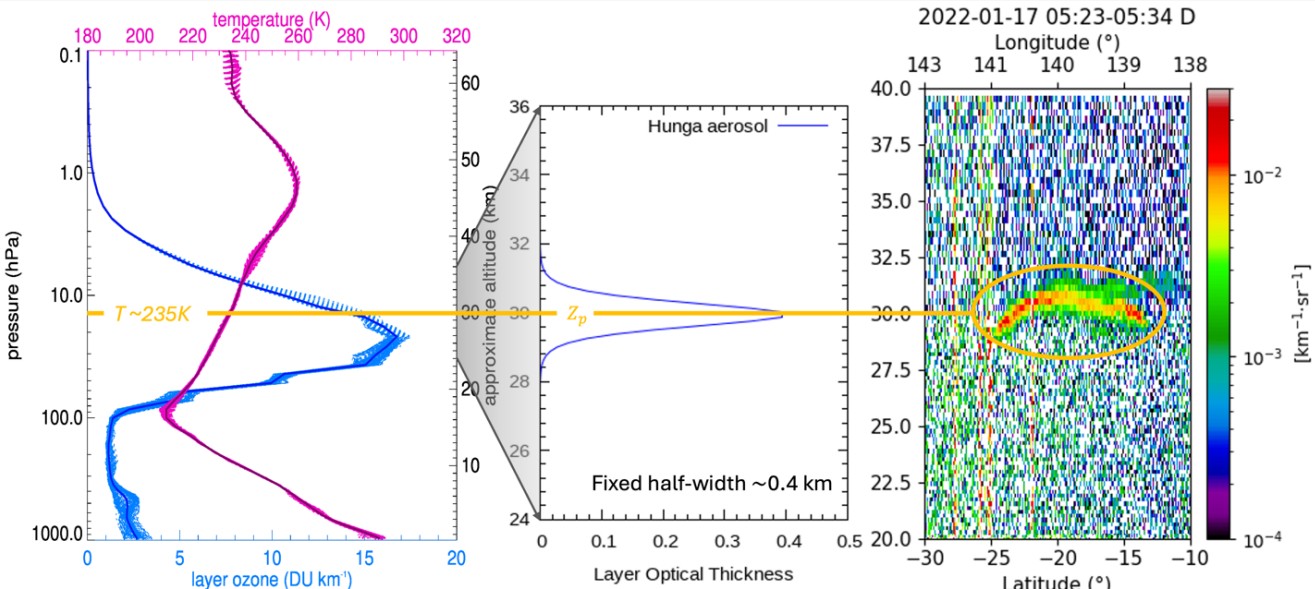

**Figure 6: (Left) Ozone and temperature profiles from the reprocessed M2-SCREAM reanalysis data on January 17, 2022, at 12 UTC. (Center) Example of a modelled aerosol profile assuming a Gaussian-like shape with fixed half-width ~0.41 km (see Appendix A). (Right) Total attenuated backscatter at 532 nm measured by CALIOP at ~5:29 UTC on January 17, 2022.**


### *Optical Properties*

The Mie code is an integral part of the forward model; we use the code to generate aerosol optical properties in the UV range, based on microphysical inputs typical for stratospheric sulfuric acid solution spherical droplets (Palmer and Williams, 1975; Beyer et al., 1996). These inputs include laboratory measurements of complex index of refraction at a reference wavelength

of 312 nm, for a binary sulfuric acid $H_2SO_4/H_2O$ solution. We have chosen two values for the real part of the refractive index ($n_r$ =1.39 and $n_r$ =1.47), which represent the lower and upper limits of the laboratory measurements (Beyer et al., 1996) which correspond to low and high measured concentrations of the binary water-sulfuric acid solutions at ~30wt% and ~ 80wt% (Beyer et al., 1996, Myhre et al., 1998). We assumed these values to be spectrally unvarying. The imaginary part of the refractive index can be neglected in the UV, visible and near infrared wavelengths (Beyer et al., 1996); this was set to a value

of $10^{-4}$ throughout.

Besides complex refractive index values $\{n_r, n_i\} = \{1.39, 10^{-4}\}$ and $\{1.47, 10^{-4}\}$, two parameters — the fine mode radius ($R_g \sim 0.14 \ \mu m$) and the standard deviation ($S_g = 1.545 \ \mu m$) — were chosen to characterize the unimodal lognormal particle number size distribution of the Hunga aerosol; these parameters were taken from the Lucinda AERONET inversions during

the Hunga cloud overpass (Boichu et al., 2023). An initial call to the Mie program is required to generate the extinction coefficient $Q_{ext}(\lambda_0)$ at reference wavelength $\lambda_0 = 312$ nm, and this is followed by more calls to the Mie code at every spectrum wavelength to generate the full set of aerosol optical properties (spectral extinction, single scattering albedo, elements



of the scattering matrix) required as input to the VLIDORT optical setup. The deployment of these Mie properties in the VLIDORT setup is discussed in Appendix B.


Ozone cross-sections are taken from laboratory measurements (Brion et al., 1993). The original data are pre-convolved with pixel-specific spectral response functions and then spline-interpolated to TROPOMI wavelengths. $SO_2$ cross-sections are taken from Bogumil et al. (2003) and are also pre-convolved and interpolated. Both cross-section data sets have quadratic-parameterized temperature dependencies based on re-processed M2-SCREAM assimilated temperature profiles (Fig. 6, left).

Rayleigh scattering cross-sections and depolarization ratios are taken from a standard source (Bodhaine et al., 1999).

*Radiative Transfer Aspects*

In the band 1 UV spectral region below 300 nm, single scattering (SS) dominates the RT for an aerosol-free stratosphere, with light penetration depths related to the wavelength-dependent ozone absorption peaks. With high-altitude Hunga aerosols

present, multiple scattering (MS) becomes more important, and it is necessary to run VLIDORT in full scattering mode (SS + MS). The number of discrete ordinates is set at 8 in the polar angle half space; we have found that this is sufficient for treating the Hunga aerosol scattering accurately, provided the delta-M scaling approximation is in force. VLIDORT is run in linear polarization mode (Stokes-vector components $I, Q\ U$); circular polarization is neglected.

**3.5 Validation with synthetic data**

We calculated Hunga BUV synthetic radiances using the NASA OMI spectral simulator software, based on geophysical conditions for January 17, 2022. Simulations were performed with and without aerosols to develop synthetic radiance ratios, which were then used as input to the Hunga inversion tool, the purpose being to evaluate the impact of changing the ozone profile and the Hunga aerosol layer height and AOD. These tests helped to develop confidence in the forward and inverse

models used in the real Hunga retrievals.

In addition, a  number of tests were carried out to obtain a sense of the retrieval's sensitivity to (1) the half width at half maximum (HWHM) of the ascribed aerosol plume profile, (2) profiles of atmospheric density and ozone, (3) parameters governing the aerosol particle size distribution [e.g. mode fraction (assuming a bimodal PSD), mode radius, refractive index

(real & imaginary parts)], (4) the spectral window chosen for retrieval, and (5) the influence of the initial state vector guess on retrieval convergence.  These tests were used as a guide to aspects of the retrieval which demanded further attention during the process of refining the retrieval software.



## 4. Hunga Aerosol Retrieval Results

To facilitate Hunga stratospheric aerosol column mass estimation, we have selected TROPOMI measurements from 17 January
2022, taken at ~1:30 pm local time (03:30 UTC), because by that time (~47 hours after the eruption) the stratospheric
$SO_2$/sulfate clouds have completely separated from the ash and ice (see Fig. 1c from Sellitto et al., 2022), but were still above
the ozone density peak at ~25 km. The absence of UV-absorbing ash is confirmed by low values of the TROPOMI-derived
UV absorbing aerosol index. As noted in section 2.1, Hunga plume pixels have been discriminated using a cloud screening
index (CSI) value greater than $> 1.1$, where CSI is the BUV radiance ratio with the background orbit (22085) at 296 nm (see
Appendix C).

### 4.1 Aerosol Peak Height Retrievals

Hunga aerosol peak heights $Z_p$ were retrieved using both lower ($n_r = 1.39$) and upper ($n_r = 1.47$) bounds of the real part of the
refractive index; these values provide an uncertainty range for Hunga $Z_p$ retrievals. Figure 7 (top) shows $Z_p$ retrieved using
the upper-bound index ($n_r = 1.47$). The highest values of $Z_p > 30$ km were retrieved in the western part of the plume from orbit
22087. The lower $Z_p$ values (~25 km) were retrieved in the eastern part from orbit 22086. This difference is largely explained
by the strong stratospheric easterly wind gradient in the 20–40 km altitude range. This interpretation is supported by the
trajectory analysis presented in Appendix D, which uses assimilated wind data from the MERRA-2 reanalysis. The wind profile
is such that the stratospheric part of the volcanic cloud moves westwards, while tropospheric parts of the cloud move eastwards.
The higher the altitude, the stronger the easterly winds, so the highest parts of the cloud at $Z_p$ ~30 km are advected westward
more quickly than the lower parts (Sadeghi et al., 2025). CALIOP profiles (Fig. 7, top) confirm the TROPOMI-retrieved Hunga
plume heights.

We compared the TROPOMI aerosol peak heights $Z_p$ retrieved from orbit 22086 at ~3:30 UTC and orbit 22087 at ~5 UTC
with the CALIOP daytime overpass at 5:27–5:29 UTC and later nighttime overpass at ~16:16 UTC (Fig. 7, top).  Examining
the daytime CALIOP data, we see that the average height of the aerosol cloud is close to 31 km. The $Z_p$ heights retrieved from
TROPOMI orbit 22087, ~30 minutes prior to the CALIOP observations, match within ~1 kilometer or ~3%. A second
validation was obtained with the CALIOP nighttime overpass at 16:16 UTC (solid magenta line), where matchup between
CALIOP and TROPOMI pixels within the Hunga plume (shown with the dashed magenta line) can be achieved with 13-hour
back trajectories (Appendix D).


TROPOMI $Z_p$ ~30 km over the northeast part of Australia agrees with the geometric top height retrievals from the Multi-angle
Imaging Spectro-Radiometer (MISR) aboard NASA's Terra satellite. On January 17, 2022, MISR observed the Hunga aerosol
plume off the northeast coast of Australia at ~00:25 UTC, with retrieved values of 27–30+ km ASL (30 km is the maximum
allowed retrieval height in the MINX (MISR INteractive EXplorer) stereo-height retrieval - see Figure 4 in Kahn et al., (2024).




Figure 7 (bottom left) shows the absolute difference in $Z_p$ retrieved using the lower-bound index ($n_r$=1.39) relative to $n_r$=1.47 scenario. Notably, in the central dense part of the plume, the differences are below ± 1 km (shaded in gray), indicating low sensitivity to the refractive index assumptions. However, localized differences of up to ±2 km were observed in the eastern part of the plume above the Coral Sea. These discrepancies may be due to the low altitude of the Eastern part of the plume, close to the assumed plume boundary (24 km). Larger $Z_p$ differences up to ~4 km were found in the western part of the cloud with low retrieved aerosol optical depth (Figure 8). Overall, differences in $Z_p$ values retrieved using extreme refractive index assumptions fall within the expected range.

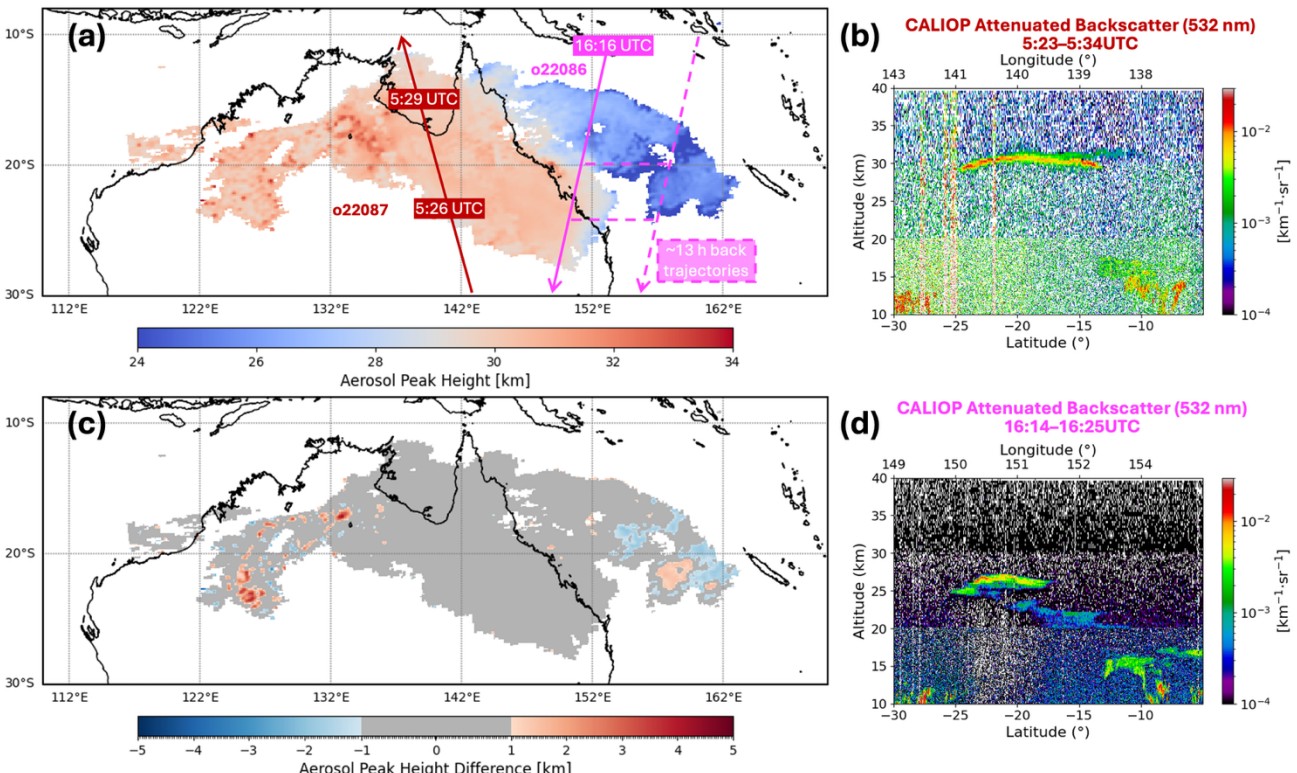

**Figure 7: (a) Retrieved aerosol plume peak height $Z_p$ [km] assuming upper limit refractive index $n_r$ = 1.47, from TROPOMI orbits 22086 (~3:20 UTC) and 22087 (~5 UTC) on January 17, 2022. (b) the CALIOP attenuated backscatter during daytime (~5:26–5:29 UTC) with the ground track shown in (a) with a solid red line. (d) same as (b) but for a nighttime track (~16:16–16:18 UTC), which is shown with a thick magenta solid line in (a). The dashed magenta line shows a back-trajectory matchup between night-time CALIOP measurements and daytime TROPOMI Hunga aerosol retrievals.**
**(c) Absolute difference in retrieved $Z_p$ assuming low ($n_r$ =1.39) and high ($n_r$ =1.47) refractive index scenarios.**



### 4.2 Aerosol Optical Depth Retrievals

To assess the sensitivity of the AOD retrieval to the assumed real part of the aerosol refractive index, we performed retrievals using two representative values of the $n_r$ (1.39 and 1.47), which span the plausible range for sulfuric acid aerosols. Figure 8 shows TROPOMI-retrieved AOD at reference wavelength 312 nm for the upper limit of the refractive index ($n_r = 1.47$) and the percentage difference in results between the two $n_r$ scenarios. The highest AOD values (up to ~5.0) were retrieved over the Coral Sea, where the densest portion of the volcanic plume was concentrated at lower plume altitudes around 25 km (see Fig.

8, top). As the plume was transported westward across northern Australia, a secondary maximum in aerosol density was located over Western Queensland, with AOD values as high as 3. Further westward, AOD values gradually decreased over Northeast Australia coinciding with a higher $Z_p$ values (> 30 km).

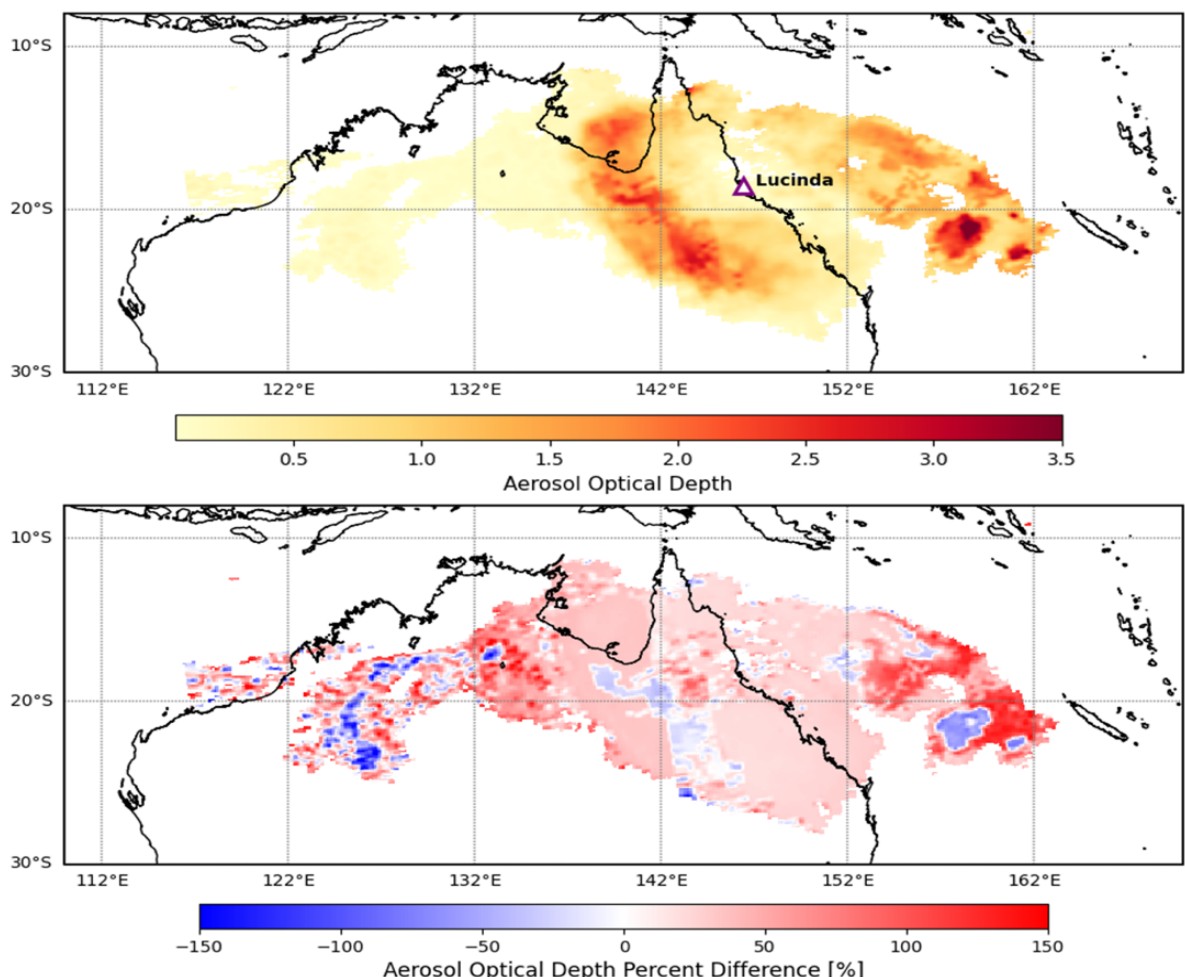

**Figure 8: (Top) Retrieved TROPOMI Aerosol Optical Depth assuming $n_r = 1.47$ at 312 nm for orbits 22086 and 22087. The location of the Lucinda AERONET site is marked with a purple triangle. (Bottom) Percentage difference in AOD values between two refractive index scenarios (1.39, 1.47) calculated relative to the $n_r = 1.47$ case.**



The comparison (Fig. 8, bottom) shows that retrieved AODs for low aqueous acid solution concentration ($n_r$ = 1.39) are
generally higher than those AODs for high solution concentration ($n_r$ = 1.47), with a mean percent difference of ~30%; this
provides an estimate of the AOD retrieval error associated with uncertainties in the refractive index assumptions. The largest
differences were found in regions with optically thick plumes, particularly over the Coral Sea, where the aerosol peak height
was close to the assumed low limit of $Z_p$ ~24 km — that is, close to the ozone density peak.

To validate our TROPOMI AOD retrievals, we have compared them with AERONET direct-sun AOD measurements (Holben
et al., 1998) from the Lucinda coastal site (18.5198°S; 146.3861°E; elevation: 8.0 m) during the Hunga plume overpass from
~21 UTC on January 16 to 03 UTC on January 17 (Fig. 9). The Lucinda site is located ~6 km offshore in the tropical coastal
waters of the Great Barrier Reef, and background AOD at this site is typically very small. Based on AERONET values of
aerosol microphysical parameters, we used the Mie code to convert our AOD values at 312 nm to corresponding quantities at
412 nm; this is the shortest AOD wavelength for AERONET measurements at Lucinda (Fig. 9, right). We also subtracted
tropospheric AOD contributions of ~0.1, as measured by AERONET previous to the Hunga plume overpass.

Using the NASA Goddard trajectory model (see Appendix D), we calculated the backward movement of air parcels starting
from TROPOMI AOD retrievals from orbit 22087 (overpass at 05:00 UTC) and from 22086 (overpass at 03:15 UTC). We
averaged all TROPOMI AOD retrievals from those parcels that pass within 10 km of the Lucinda site and compared them with
the AERONET AOD measurements averaged over a 15-minute interval (Fig. 9, left). We see that average retrieved AOD
values show good qualitative agreement with the AERONET AOD measurements. Both TROPOMI and AERONET AODs
reached a maximum (over 2) during the local-time morning hours (21:45–23:00 on Jan 16 UTC) and dropped to ~0.5 after the
main part of the Hunga aerosol cloud passed over the station.


TROPOMI AOD retrievals also agree qualitatively with MISR-retrieved $AOD_{558}$ ~ $0.7 \pm 0.2$ ($1\sigma$) (Kahn et al., 2024),
accounting for spectral differences in the extinction (Fig. 9, right).





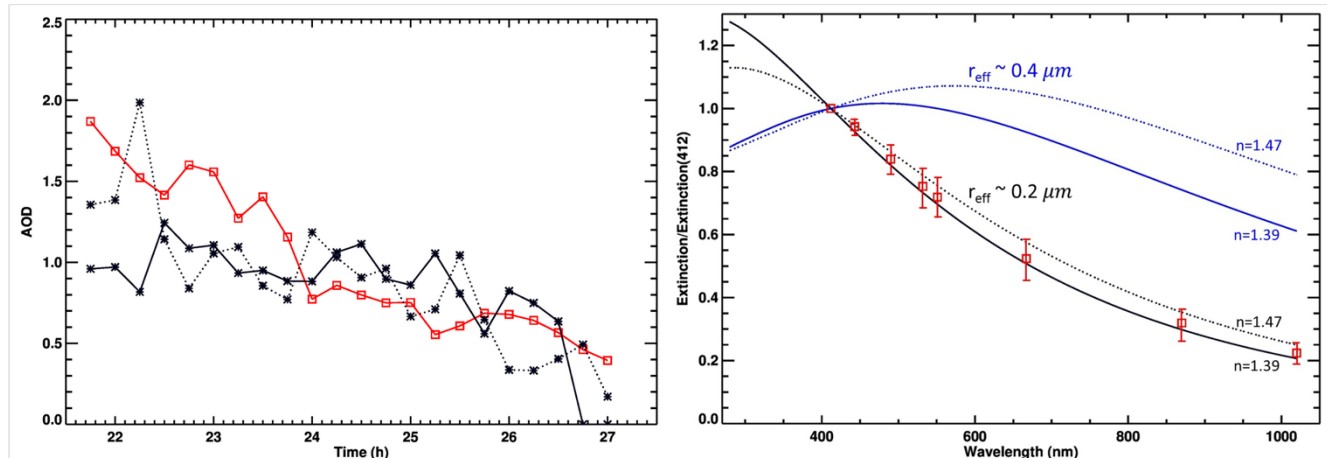

**Figure 9: (Left)** The red line shows 15-minute averages of the AERONET direct sun AOD measurements at the shortest 412nm wavelength ($AOD_{412}$). The measurements at Lucinda site started at 21:45 UTC on 16 January 2022 16 (local morning time) and continued until ~ 03:00 UTC on January 17, 2022. The horizontal scale is given in hours elapsed since 00:00 on 16 January, UTC, i.e., 03:00 on 17 January (UTC). We subtracted a background tropospheric AOD of 0.1 retrieved from AERONET measurements on the previous day. The black solid (dotted) line represents average TROPOMI AOD retrievals for all pixels that pass within 10 km of the Lucinda site, assuming $n_r$=1.47 ($n_r$=1.39) for the same time intervals. Our AOD retrievals at 312 nm were adjusted to values at 412 nm using Mie extinction spectral dependence shown on the right.

**(Right)** The red squares show the average spectral dependence of the AERONET/Lucinda Hunga AOD measurements, normalized to 412 nm. These data were obtained by averaging 40 AERONET measurements taken during the passage of the Hunga cloud from 21.61 h on January 16 to 0.72 h on January 17 (UTC). The 3× standard deviation of the ratios $AOD/AOD_{412}$ is shown as a bar ($\pm 3\sigma$). The black solid (dotted) curve shows the theoretical Extinction ratio from Mie calculations assuming $n_r$=1.39 ($n_r$=1.47) and effective radius $r_{eff}$ ~ 0.2μm. The blue curves show similar extinction ratios using a larger effective radius $r_{eff}$ ~ 0.4 μm retrieved in March 2022 by the solar occultation SAGE-III instrument aboard the International Space Station (Duchamp et al., 2023).

AOD uncertainties $\varepsilon$AOD were estimated using two different approaches: (1) $\varepsilon$AOD as returned by the Hunga retrieval tool ($n_r$=1.39 and 1.47), which is a random error; (2) a systematic error estimation extending the fitting window to full window (289–310 nm; Fig. 3). We note the following:

(1)   Our estimates of the retrieval $\varepsilon$AOD uncertainties were initially based on measurement SNR but were further refined by the incorporation of additional contributions derived from chi-square diagnostics accounting for discrepancies
between measured and simulated radiance ratios (see Section 3.3). Figure 10 shows the normalized probability density function of $\varepsilon$AOD; this curve has a long non-Gaussian tail. Clearly, we cannot use standard Gaussian diagnostics to express uncertainties.

For the case with $n_r$ = 1.47, approximately 93% of pixels fall within $\varepsilon$AOD < 0.3, and the averaged percentage error over the Hunga plume is ~16% for pixels with AOD > 0.2. In comparison, for the $n_r$ = 1.39 case, about 90% of pixels
fall within the same threshold, and the averaged percentage error is slightly lower at ~14%; this is due to the systematically higher retrieved AOD values found with this $n_r$ value. We interpret the AOD retrieval uncertainties as bounded by the range implied by these two cases—with ~14–16% representing a reasonable lower and upper limit for the percent error over the Hunga plume based on forward model diagnostics.





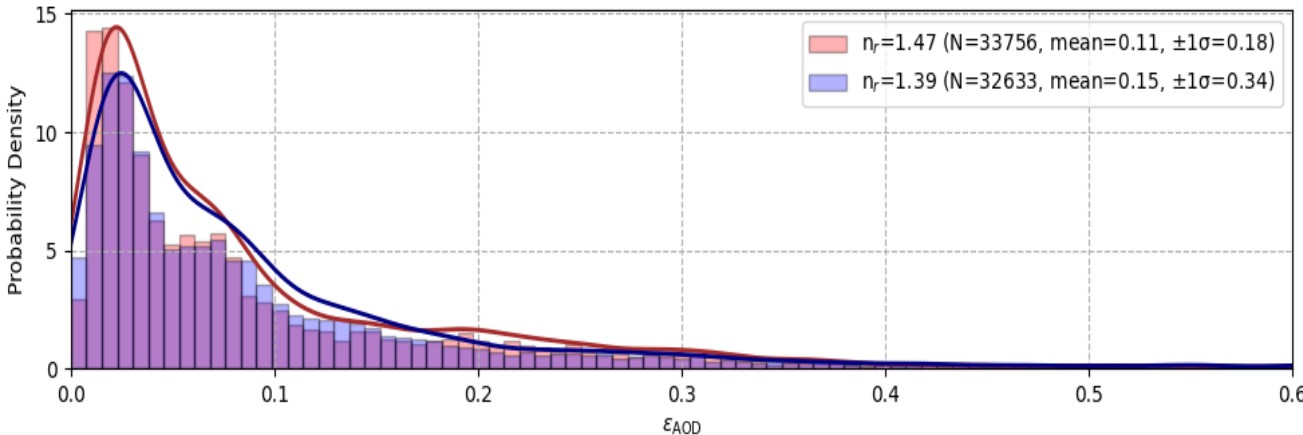

**Figure 10: Normalized probability density function of retrieved Hunga εAOD for the Hunga Plume on January 17, 2022. Results are shown for two different refractive index values: $n_r$ = 1.47 (red) and $n_r$ = 1.39 (blue). The number of valid retrievals (N) and corresponding mean ±1σ are indicated in the legend.**

(2) To confirm this upper limit of the εAOD, we repeated two TROPOMI retrievals using the 'full' fitting window (289–310 nm), assuming $n_r$ = 1.47. With this fitting window, we obtained AOD values ~16% lower than those obtained with the 'short' window. This reduction is possibly due to increased sensitivity to tropospheric clouds and gas-phase absorbers such as $SO_2$. These comparisons indicate the importance of varying the spectral fitting window to estimate total physical uncertainty of our Hunga aerosol retrievals. Using a longer spectral fitting window would permit retrieval of additional aerosol parameters (e.g., effective radius) or gases (e.g., $O_3$, $SO_2$), but would require a more complex forward RT model (e.g., including tropospheric cloud correction).

The overall εAOD is estimated using the forward model error (14–16%), and the spectral fitting window variation (−16%). Based on these estimates, we adopt ±16% as the upper limit of the εAOD percent uncertainty, considering both retrieval sensitivity and inter-sensor comparisons. This provides an uncertainty for Hunga aerosol mass retrievals.

Furthermore, we carried out an inter-sensor comparison against the OMPS-NP-based Hunga AOD retrieval; this was conducted using the same 'short' spectral fitting window (289–296 nm) and assimilated $O_3$ profiles (see Appendix E). The retrieved OMPS-NP AOD values were approximately 20% higher than those from our TROPOMI retrievals collocated within six OMPS-NP pixels over Northeast Australia.





### 4.3 Aerosol Mass Retrievals

To convert the retrieved AOD to aerosol column mass $m_{aer}$ [g/m²], we need to know the particle size distribution, characterized by effective radius $r_{eff}$ and extinction efficiency $Q_{ext} = \frac{<E>}{<G>}$ , where $<E>$ and $<G>$ are average extinction and geometric cross-sections, as well as mass density $\rho$ (Krotkov et al., 1999; Sellitto et al., 2024):

$$m_{aer} = \frac{4}{3} \frac{\rho r_{eff}}{Q_{ext}} AOD \tag{4.1}$$

We assume the AERONET-retrieved fine-mode effective radius $r_{eff} \sim 0.22 \,\mu m$, and an upper limit of the mass density $\rho \sim 1.75$ [g/cm³] from the laboratory-measured density of the 76.5 wt% sulfuric acid solution at Hunga plume temperature (Beyer et al., 1996, Myhre et al., 1998). These assumptions, together with an upper limit of the refractive index of $n_r = 1.47$, are used to produce the spatial distribution of aerosol column mass $m_{aer}$ shown in Fig. 11. The column mass spatial distribution has similar features to those in the AOD map (Fig. 8, top). Using a BUV radiance ratio filter at 296 nm (CSI > 1.1) to define the plume, we estimate a total plume area $A_{total} \sim 4 \times 10^6$ km², and a corresponding "wet" aerosol mass of $\sim 0.47$ Tg. An $m_{aer}$ up to $\sim 0.8$ g/m² was found over the densest part of the plume over the Coral Sea, as discussed in Section 4.2, since the aerosol column mass values are proportional to the AOD. Over northeast Australia, $m_{aer}$ increased in value to $\sim 0.5$ g/m², then decreased to $m_{aer} \sim 0.05$ g/m² over the northwestern part of the continent.

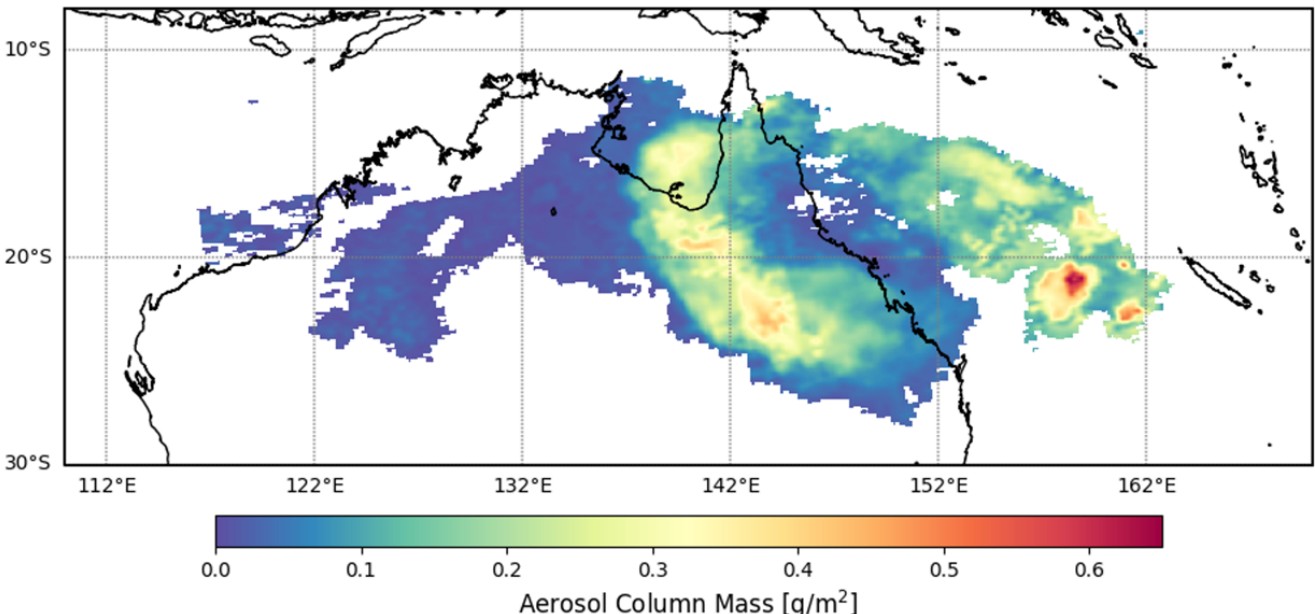

**Figure 11: Hunga aerosol column mass on January 17, 2022, assuming aqueous acid solution mass fraction 0.765, corresponding density $\rho \sim 1.75$ [g/cm³] (Myhre et al., 1998), refractive index, $n_r = 1.47$ (Beyer et al., 1996), and an AERONET retrieved fine-mode effective radius $r_{eff} \sim 0.22 \mu m$.**



The aerosol mass density ($\rho$) and the real part of the refractive index ($n_r$) are both linked to the assumed sulfuric acid
concentration in the aerosol solution droplets. Higher sulfuric acid content results in both higher $n_r$ and $\rho$, and vice versa. To
quantify the total "wet" aerosol mass $M_{aer}$, we integrated the retrieved $m_{aer}$ using two limiting ($n_r$, $\rho$) pairs:

- for $n_r = 1.47$ and $\rho = 1.75$ g/cm³ (76.5 wt%), the integrated total aerosol mass $M_{aer}$ is ~0.47 Tg.
- for $n_r = 1.39$ and $\rho = 1.25$ g/cm³ (29.1 wt%), the integrated total aerosol mass $M_{aer}$ is ~0.50 Tg.

These results for $M_{aer}$ are close; the increase in AOD and opposite decrease $Q_{ext} = \frac{<E>}{<G>}$ values are responsible when going
from $n_r = 1.47$ to $n_r = 1.39$ in eq. (4.1). They represent the lower and upper bounds of the retrieved "wet" aerosol mass (0.47–
0.50 Tg), reflecting the impact of microphysical assumptions on the retrieval. We provide a representative estimate of the total
"wet" aerosol mass ~0.5 ± 0.05 Tg; this ~10% uncertainty reflects the AOD retrieval uncertainties discussed in Section 4.2.

## 4.4 Equivalent Sulfur (S) Mass Estimate

The "wet" aerosol mass $M_{aer}$ retrieved in our study remains nearly constant (~0.5 ± 0.05 Tg) for a broad range of sulfuric acid
solution densities. We can further estimate the ambient solution density using reported Hunga sulfur (S) emissions $M_{S_0}$ ~ 0.27
Tg S (i.e., half of ~0.54 Tg SO₂ reported in Carn et al., 2022) and a constant SO₂-to-sulfate conversion rate of $\tau \approx 6$ days (e-
folding time):

$$M_{S,expected} = M_{S_0} \times [1 - e^{(-\frac{t}{\tau})}] \qquad (4.2)$$

Using Eqn. 4.2, we estimated the expected sulfur mass converted to aerosols after ~47 hours to be $M_{S,expected}$ ~0.077 TgS.
The sulfur mass in the Hunga aerosol solutions $M_{S,aer}$ can be estimated from the retrieved "wet" aerosol mass, $M_{aer}$ using the
following equation (4.3):

$$M_{S,aer} = M_{aer} w \frac{MW_S}{MW_{H_2SO_4}} \qquad (4.3)$$

Here, $w$ is the mass fraction of sulfuric acid in the solution (29.1 w% - 76.5wt%) and $\frac{MW_S}{MW_{H2SO4}}$ is the molecular mass ratio
between S (~32 g/mol) and H₂SO₄ (~98 g/mol), which equals 0.3265. Applying this approach, the total aerosol-phase equivalent
S mass is $M_{S,aer}$ ~0.047–0.116 TgS. Comparing this range with the expected S mass already oxidized to aerosol, $M_{S,expected}$
~0.077 TgS, we estimate the ambient Hunga aerosol solution concentration to be $w_{ambient}$~50 wt%.

To assess the total S burden in the Hunga plume, we calculated the gas-phase sulfur mass ($M_{S,gas} = 0.135$ TgS) within the
Hunga stratospheric aerosol plume (CSI > 1.1 and $Z_p$ > 24 km), based on operational TROPOMI SO₂ VCD retrievals (Theys
et al., 2017). The total S mass (gas plus aerosol) is calculated to be ~0.212 TgS. This value should be interpreted as a lower
bound, given that (1) we retrieve only aerosol mass above the ozone density peak (>24 km), and (2) gas-phase SO₂ mass is
limited to pixels collocated with the aerosol plume.



**4.5 Comparison with Infrared Measurements**

Sellitto et al. (2024) reported retrievals of $SO_2$ and sulfate aerosol mass in the Hunga plume based on mid-IR IASI measurements. Their study noted that $SO_2$ and sulfate aerosol have overlapping spectral signatures in the IR, which can lead to large uncertainties in the co-retrieval of these species in volcanic plumes; however, the potential impact of collocated water vapor on the IR retrievals was not addressed. In their paper, the equation used to derive sulfate mass from IASI measurements of mid-IR AOD is identical to that used here (Eq. 4.1) but is based on the measured mid-IR AOD and average extinction efficiency ($Q_{ext}$) calculated at mid-IR wavelengths (~8.5 µm). Sellitto et al. (2024) also assumed a sulfate aerosol mass density of 1.75 [g cm$^{-3}$] which corresponds to the upper limit of the sulfuric solution concentration 76.5 wt% (Myhre et al., 1998). Additional uncertainties arise from the range of possible particle size distributions ($r_{eff}$ ~0.25–0.45 µm) in the Hunga aerosol plume.

These authors report a maximum Hunga sulfate aerosol mass loading of $1.6 \pm 0.5$ Tg, but the peak loading was measured later in the year (August-September 2022). On January 17, 2022, the IASI-derived sulfate aerosol mass reported in Sellitto et al. (2024) is ~0.2 Tg, with a large uncertainty (estimates range up to ~0.8 Tg). Our UV-based "wet" aerosol mass of $M_{aer}$ ~0.5 Tg is thus broadly consistent with these IR retrievals, given the uncertainties on the assumed particle size distribution. In contrast, a larger discrepancy is apparent in the $SO_2$ retrievals: on January 17, the IASI-based $SO_2$ mass is ~0.75 Tg (range: ~0.4–1.1 Tg), compared to a total BUV TROPOMI $SO_2$ mass of ~0.4 Tg (note that this is the total retrieved $SO_2$ mass on January 17, not the $SO_2$ collocated with the Hunga aerosol plume discussed in section 4.4). Furthermore, the IASI-based $SO_2$ mass reached a maximum of ~1 Tg (range: ~0.7–1.2 Tg) on January 19, 2022 (Sellitto et al., 2024), whereas the BUV $SO_2$ mass was observed to decrease after January 17 (e.g., Carn et al., 2022).

Sadeghi et al. (2025) also retrieved $SO_2$ mass using CrIS IR measurements in combination with the VOLCAT (VOLcanic Cloud Analysis Toolkit) framework. This work employed IR retrievals from the NASA-NOAA's Joint Polar Satellite System (JPSS) satellites and HYSPLIT-based trajectory analysis to assess $SO_2$ transport and decay patterns. On January 16, 2022, Sadeghi et al. (2025) reported a CrIS-derived $SO_2$ mass of ~0.4 Tg, which is consistent with the BUV $SO_2$ mass reported in Carn et al. (2022).

Reasons for these discrepancies are difficult to sort out. It is possible that the BUV $SO_2$ measurements were more strongly impacted by the presence of optically thick aerosol; we also propose that the impact of Hunga water vapor on the IR $SO_2$ and sulfate mass retrievals may also merit further consideration.



**5. Summary and Conclusions**

The January 15, 2022 eruption of the submarine Hunga volcano was a unique volcanic event in the ~50 years since the
beginning of the satellite remote sensing era. This powerful eruption was the largest since Pinatubo in 1991, but unlike the
SO$_2$-rich Pinatubo emissions, Hunga injected a volcanic plume dominated by water vapor, with relatively low SO$_2$ content, to
altitudes as high as the lower mesosphere. Although the Hunga eruption has been studied intensively from a number of remote
sensing perspectives, in this work we have presented a novel retrieval of aerosol mass and layer height using BUV
measurements from the S5P/TROPOMI instrument on January 17, 2022. These unique BUV retrievals allow us to detect and
characterize the mid-stratospheric Hunga aerosol plume that moved across the Southwest Pacific and Australia about 47 hours
after the January 15 eruption. This study demonstrates for the first time that BUV radiance measurements can be used to
retrieve mid-stratospheric AOD, $Z_p$ and aerosol mass above the ozone density peak ~25 km, following a major volcanic
eruption.

Our algorithm simultaneously retrieves AOD and aerosol peak height using radiance ratios (TROPOMI measurements in the
presence of the Hunga aerosol plume divided by background aerosol-free measurements from a preceding orbit) in the 289–
296 nm spectral fitting window. To identify Hunga aerosol plumes and exclude tropospheric clouds, we determined empirically
the threshold of the BUV radiance ratio at 296 nm (CSI, cloud screening index), restricting retrievals to TROPOMI pixels with
CSI > 1.1. Our work also provides an explanation of the observed anomalies in total ozone column (TOC) BUV retrievals in
the presence of the Hunga aerosol plume, when enhanced aerosol scattering increases BUV radiances, leading to erroneously
low anomalies in ozone retrievals. In this study, O$_3$ profile data from reprocessed M2-SCREAM reanalysis (excluding the
anomalous TOC retrievals in the assimilation) were used to properly account for strong ozone absorption effects. For the
Hunga aerosol retrieval, Mie calculations were performed to derive aerosol spectral optical properties in the UV range, based
on microphysical inputs typical for H$_2$SO$_4$/H$_2$O solutions at stratospheric temperatures and pressures. We have developed a
new forward-modeling tool based on the VLIDORT vector radiative transfer code, with the retrieval algorithm state vector
comprising only two aerosol parameters: the AOD at 312 nm and the aerosol layer peak height. The retrieval inverse model
employs a modified Levenberg-Marquardt iterative least-squares inversion.

Our aerosol retrievals were validated using satellite-based lidar (CALIOP), Multi-angle Imaging Spectro-Radiometer (MISR)
and ground-based AERONET direct-sun AOD measurements using trajectory modeling. Additionally, trajectory modeling
showed that air parcels back-propagated from TROPOMI retrievals were consistent with the stratospheric wind and transport
pathways of the Hunga plume.

We used the retrieved AOD to estimate a Hunga total "wet" aerosol mass of ~0.5 ± 0.05 Tg. Assuming a 50% sulfuric acid in
water solution, $M_{S,aer} \sim 0.077 \, TgS$, which represents the portion of emitted SO$_2$ already oxidized into aerosols by 17 January



2022. While the retrieved $M_{S,aer}$ after 47 hours of transport provides insight into the SO₂-to-sulfate conversion rate, a more comprehensive assessment would require tracking this process over a longer timescale using additional aerosol mass retrievals. A direct comparison between SO₂ decline and sulfate aerosol mass increase over time, coupled with chemical modelling, could allow for a more precise quantification of SO₂ decay rates in the stratosphere—refining our understanding of sulfur removal

mechanisms and the relationship between gas-phase sulfur loss and aerosol formation.

## Appendix A. Aerosol Plume Parameterization

The treatment here follows that in (Spurr and Christi, 2014). We use the same pseudo-Gaussian plume parameterization scheme

for aerosols and for the other trace gases (SO₂, O₃); the exposition here is given just for aerosols but applies equally to the two trace species. The aerosol plume is characterized by three parameters $\{A_0, z_p, h_w\}$: $A_0$ is the plume total optical depth at a fixed reference wavelength $\lambda_{ref}$ (312 nm), $z_p$ is the plume peak height in [km], and $h_w$ is the HWHM in [km] of the plume distribution. We retrieve the first two of these parameters; the state vector is $\mathbf{x} = \{A_0, z_p\}$. The pseudo-Gaussian plume is the aerosol optical thickness profile at 312 nm, given by:

$$\tau(z) = \Omega \frac{\exp\left[-f(z-z_p)\right]}{\left[1+\exp\left[-f(z-z_p)\right]\right]^2}. \tag{A2.1}$$

Here, $z$ is the altitude, $z_p$ is the peak height ("PKH"), $\Omega$ is a normalization constant related to total stratospheric aerosol optical thickness ("AOD") $A_0$, and $f$ is an exponential constant related to the HWHM parameter $h_w$ through $fh_w = \ln\left[3 + 2\sqrt{2}\right]$. At peak height $z = z_p$, the loading is $\tau(z_p) = \frac{1}{4}\Omega$.

We assume that the plume lies between two limiting heights $z_b$ and $z_t$. Integrating the profile between these limits yields the

total AOD:

$$A_0 = \int_{z_b}^{z_t} \tau(z)dz = \Omega\Gamma; \quad \Gamma = \frac{(Y_b-Y_t)}{(1+Y_b)(1+Y_t)}; \tag{A2.2}$$

$$Y_b = \exp\left[-f(z_b - z_p)\right]; \quad Y_t = \exp\left[-f(z_t - z_p)\right]. \tag{A2.3}$$

For a discretization of the atmosphere into vertical layers $\{z_n\}, n = 0,1,\dots N_L$, where $N_L$ is the total number of layers, the loading profile will be given by:

$$L_n = \int_{z_n}^{z_{n-1}} \tau(z)dz = \frac{A_0}{\Gamma}\Gamma_n; \quad . \tag{A2.4}$$

$$\Gamma_n = \frac{(Y_n-Y_{n-1})}{(1+Y_n)(1+Y_{n-1})}; \quad Y_n = \exp\left[-f(z_n - z_p)\right]. \tag{A2.5}$$

Here we have used Eq. (A2.2) to show that each layer amount $L_n$ is directly proportional to $A_0$. The forward model radiative transfer calculation using VLIDORT requires Jacobians with respect to $A_0$ and $z_p$, plus $h_w$ if the latter is to be included in the retrieval or is to be considered as a model parameter error in the retrieval. We require partial derivatives of the loading profile

with respect to these parameters. Explicit differentiation of Eq. (A2.4) gives:



$$\frac{\partial L_n}{\partial A_0} = \frac{\Gamma_n}{\Gamma}; \quad \frac{\partial L_n}{\partial z_p} = \frac{1}{\Gamma} \cdot \left[\frac{\partial \Gamma_n}{\partial z_p} - L_n \frac{\partial \Gamma}{\partial z_p}\right]; \quad \frac{\partial L_n}{\partial h_w} = \frac{1}{\Gamma} \cdot \left[\frac{\partial \Gamma_n}{\partial h_w} - L_n \frac{\partial \Gamma}{\partial h_w}\right]. \tag{A2.6}$$

The $A_0$ derivative is trivial. Derivatives with respect to $z_p$ are harder to establish; after some algebra, we find the auxiliary
derivatives of $\Gamma$ and $\Gamma_n$ through:

$$\frac{\partial \Gamma_n}{\partial z_p} = f\Gamma_n \frac{(1-Y_n Y_{n-1})}{(1+Y_n)(1+Y_{n-1})}; \qquad \frac{\partial \Gamma}{\partial z_p} = f\Gamma \frac{(1-Y_b Y_t)}{(1+Y_b)(1+Y_t)}. \tag{A2.7}$$

Similarly, the auxiliary derivative $\Gamma_n$ with respect to $h_w$ is given by:

$$\frac{\partial \Gamma_n}{\partial h_w} = -\frac{C}{f^2}\Gamma_n \left[z_p - \frac{(z_n Y_n - z_{n-1} Y_{n-1})}{(Y_n - Y_{n-1})} + \frac{(z_n - z_p)Y_n}{(1+Y_n)} + \frac{(z_{n-1} - z_p)Y_{n-1}}{(1+Y_{n-1})}\right]. \tag{A2.8}$$

A similar expression holds for the derivative of $\Gamma$ with respect to $h_w$, but with $Y_b$ and $z_b$ replacing $Y_n$ and $z_n$, and $Y_t$ and $z_t$
replacing with $Y_{n-1}$ and $z_{n-1}$. In Eq. (A2.8), the constant $C = \ln[3 + \sqrt{8}]$.

Figure A1 (top panel) illustrates three typical pseudo-Gaussian plumes, with total AOD $A_0 = 1.81$, peak height $z_p = 31.5$ km
and three different values of $h_w$ as indicated. The lower panels show the partial derivatives with respect to $A_0$ and $z_p$.

Treatment of the SO₂ trace gas profiles is similar. Plume parameters are the total column $\Omega_{SO2}$ in [DU], and the aerosol
parameters $z_p$ and $h_w$, when the plumes are positioned together and have the same shape. In this case, derivatives of the SO₂
plume profile with respect to $z_p$ and $h_w$ will then have exactly the same form as the expressions in Eqns. (A2.6) to (A2.8).



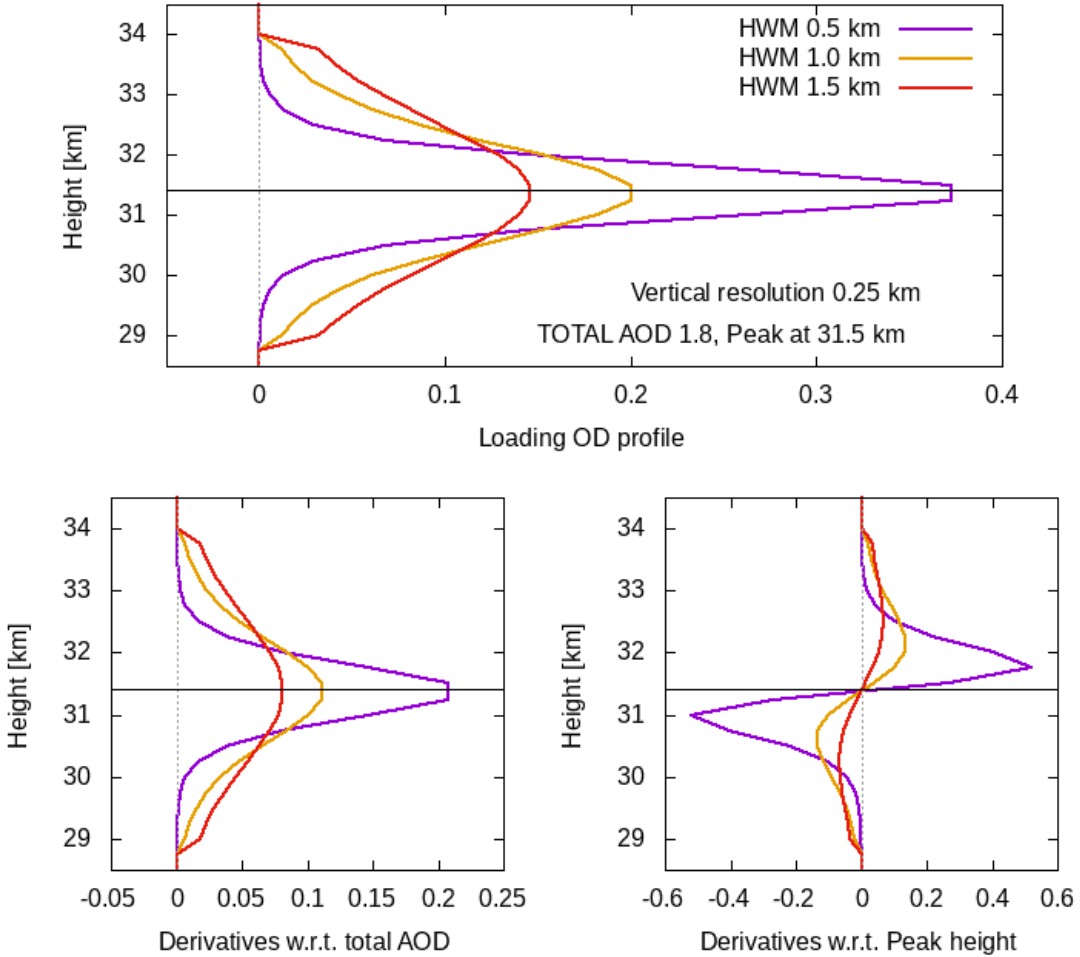

**Figure A1: (Upper panel) Pseudo-Gaussian aerosol plume profiles over the 28–35 km height range, for three different HWHM values as indicated. (Lower panels) Profile derivatives with respect to AOD and peak height.**

## Appendix B. VLIDORT and the Forward Model

VLIDORT is a discrete-ordinate polarized radiative transfer (RT) model in wide use in the remote sensing community. Single
scattering in VLIDORT is treated accurately for line-of-sight and solar paths allowing for the Earth's curvature, while the
multiple-scatter field is determined through plane-parallel scattering along with the pseudo-spherical approximation (solar
beam attenuation for a curved atmosphere). The great advantage using VLIDORT lies in its ability to return not just the
backscattered Stokes-vector radiation field, but also analytically-derived Jacobians of this field with respect to any atmospheric
or surface property.






To cover all possible retrieval trials discussed in this work and planned sequel papers, we require VLIDORT to calculate Jacobians with respect to the three aerosol parameters $\{A_0, z_p, h_w\}$, the single $SO_2$ parameter $\{\Omega_{SO2}\}$ and the three $O_3$ parameters $\{\Omega_{O3}, z_{p,O3}, h_{w,O3}\}$.

In VLIDORT, the atmosphere is taken as a series of optically uniform layers. Without loss of generality, the standard set of input optical properties (IOPs) is $\{\Delta_n, \omega_n, \mathbf{B}_{nl}\}, n = 1, \dots N_L$ , where $\Delta_n$ is the layer optical depth for extinction in layer $n$, $\omega_n$ the total single scattering albedo in that layer, and $\mathbf{B}_{nl}$ is a 4x4 matrix of spherical-function expansion coefficients that are used to develop the total scattering and phase matrices. [Scattering matrices can be specified in advance for the single-scattering calculations, as an alternative to developing them from sets of expansion coefficients]. For Jacobians, VLIDORT also requires

the set of *linearized* IOPs $\{\mathcal{V}_{nq}, \mathcal{U}_{nq}, \mathbf{Z}_{nlq}\}, n = 1, \dots N_L$ defined as the double-normalized partial derivatives of the IOPs with respect to Jacobian parameter $\xi_q$. In other words:

$$\mathcal{V}_{nq} = \frac{\xi_q}{\Delta_n}\frac{\partial \Delta_n}{\partial \xi_q}; \quad \mathcal{U}_{nq} = \frac{\xi_q}{\omega_n}\frac{\partial \omega_n}{\partial \xi_q}; \quad \mathbf{Z}_{nlq} = \frac{\xi_q}{\mathbf{B}_{nl}}\frac{\partial \mathbf{B}_{nl}}{\partial \xi_q}. \quad \text{(B2.1)}$$

Here, we determine these IOPs and associated parameter derivatives for the present application.

If the trace gas absorption optical thickness is $G_n(\lambda)$ in layer $n$, the Rayleigh scattering optical thickness $R_n(\lambda)$, and the aerosol extinction optical thickness $E_n(\lambda)$ at wavelength $\lambda$, then the IOPs in that layer are:

$$\Delta_n(\lambda) = G_n(\lambda) + R_n(\lambda) + E_n(\lambda); \quad \text{(B2.2a)}$$

$$\omega_n(\lambda) = \frac{R_n(\lambda) + a(\lambda)E_n(\lambda)}{\Delta_n(\lambda)}; \quad \text{(B2.2b)}$$

$$\mathbf{B}_{nl}(\lambda) = \frac{R_n(\lambda)\mathbf{B}_l^{(Ray)}(\lambda) + a(\lambda)E_n(\lambda)\mathbf{B}_l^{(Aer)}(\lambda)}{R_n(\lambda) + a(\lambda)E_n(\lambda)}. \quad \text{(B2.2c)}$$

Here $a(\lambda)$ is the aerosol single scatter albedo, with $\mathbf{B}_l^{(Ray)}$ and $\mathbf{B}_l^{(Aer)}$ the coefficient matrices for Rayleigh and aerosol scattering respectively.

Now the aerosol optical thickness $E_n(\lambda)$ is related to the aerosol loading profile $\{L_n\}$ at reference wavelength $\lambda_0$ through:

$$E_n(\lambda) = r(\lambda)L_n = \frac{\epsilon(\lambda)}{\epsilon(\lambda_0)}L_n. \quad \text{(B2.3)}$$

Here, $\epsilon(\lambda)$ is the coefficient for aerosol extinction at the wavelength of interest, with $\epsilon(\lambda_0)$ the extinction coefficient at reference wavelength $\lambda_0$, with $r(\lambda)$ the ratio of these two quantities.

Similarly, the trace gas absorption term (with $SO_2$ included) is

$$G_n(\lambda) = \sigma_{n,O3}(\lambda)L_{n,O3} + \sigma_{n,SO2}(\lambda)L_{n,SO2}; \quad \text{(B2.4)}$$

Here, $\{L_{n,O3}\}$ and $\{L_{n,SO2}\}$ are trace gas loading profiles, with absorption cross-sections denoted by $\sigma_{n,O3}(\lambda)$ and $\sigma_{n,SO2}(\lambda)$.



Given the aerosol loading profile $\{L_n\}$ and gas profiles $\{L_{n,O3}\}, \{L_{n,SO2}\}$, we are now in a position to derive the linearized optical properties in Eq. (B2.1) through explicit chain-rule differentiation of the results In Eq. (B2.2)-B(2.4) with respect to any of the three aerosol parameters $\{A_0, z_p, h_w\}$, the SO$_2$ parameter $\{\Omega_{SO2}\}$ or the three O$_3$ parameters $\{\Omega_{O3}, z_{p,O3}, h_{w,O3}\}$.

Dealing first with the aerosol profile $\{L_n\}$, and using the symbol $\xi$ to indicate any one of the parameters $\{A_0, z_p, h_w\}$, we find that:

$$\frac{\partial \Delta_n(\lambda)}{\partial \xi} = r(\lambda) \frac{\partial L_n}{\partial \xi}; \tag{B2.5a}$$

$$\frac{\partial \omega_n(\lambda)}{\partial \xi} = r(\lambda) \frac{\partial L_n}{\partial \xi} \cdot \left[ \frac{a(\lambda) - \omega_n(\lambda)}{\Delta_n(\lambda)} \right]; \tag{B2.5b}$$

$$\frac{\partial \mathbf{B}_{nl}(\lambda)}{\partial \xi} = a(\lambda) r(\lambda) \frac{\partial L_n}{\partial \xi} \cdot \left[ \frac{\mathbf{B}_{nl}^{(Aer)}(\lambda) - \mathbf{B}_{nl}(\lambda)}{R_n(\lambda) + a(\lambda) E_n(\lambda)} \right]. \tag{B2.5c}$$

Dealing next with the O$_3$ profile $\{L_{n,O3}\}$ and setting $\xi_{O3}$ to any of the three O$_3$ parameters $\{\Omega_{O3}, z_{p,O3}, h_{w,O3}\}$, we have:

$$\frac{\partial \Delta_n(\lambda)}{\partial \xi_{O3}} = \sigma_{n,O3}(\lambda) \frac{\partial L_{n,O3}}{\partial \xi_{O3}}; \tag{B2.6a}$$

$$\frac{\partial \omega_n(\lambda)}{\partial \xi_{O3}} = -\frac{\omega_n(\lambda)}{\Delta_n(\lambda)} \cdot \frac{\partial \Delta_n(\lambda)}{\partial \xi_{O3}}; \tag{B2.6b}$$

$$\frac{\partial \mathbf{B}_{nl}(\lambda)}{\partial \xi_{O3}} = 0. \tag{B2.6c}$$

Note that these ozone derivatives are only present for the parameterized part of the profile; outside this range they are zero.

The situation with SO$_2$ is a little more complicated. For the SO$_2$ loading parameter $\Omega_{SO2}$, the derivatives are of the same form as those in Eqn. (B2.6):

$$\frac{\partial \Delta_n(\lambda)}{\partial \Omega_{SO2}} = \sigma_{n,SO2}(\lambda) \frac{\partial L_{n,SO2}}{\partial \Omega_{SO2}}; \quad \frac{\partial \omega_n(\lambda)}{\partial \Omega_{SO2}} = -\frac{\omega_n(\lambda)}{\Delta_n(\lambda)} \cdot \frac{\partial \Delta_n(\lambda)}{\partial \Omega_{SO2}}; \quad \frac{\partial \mathbf{B}_{nl}(\lambda)}{\partial \Omega_{SO2}} = 0. \tag{B2.7}$$

If the SO$_2$ plume is coincident with the aerosol plume, then there will be additional dependencies on the parameters $\{z_p, h_w\}$. Thus we now have (in place of (B2.5)):

$$\frac{\partial \Delta_n(\lambda)}{\partial z_p} = r(\lambda) \frac{\partial L_n}{\partial z_p} + \sigma_{n,SO2}(\lambda) \frac{\partial L_{n,SO2}}{\partial z_p}; \tag{B2.8a}$$

$$\frac{\partial \omega_n(\lambda)}{\partial z_p} = \frac{1}{\Delta_n(\lambda)} \left[ a(\lambda) r(\lambda) \frac{\partial L_n}{\partial z_p} - \omega_n(\lambda) \frac{\partial \Delta_n(\lambda)}{\partial z_p} \right]; \tag{B2.8b}$$

$$\frac{\partial \mathbf{B}_{nl}(\lambda)}{\partial z_p} = a(\lambda) r(\lambda) \frac{\partial L_n}{\partial z_p} \cdot \left[ \frac{\mathbf{B}_{nl}^{(Aer)}(\lambda) - \mathbf{B}_{nl}(\lambda)}{R_n(\lambda) + a(\lambda) E_n(\lambda)} \right]. \tag{B2.8c}$$

This establishes the necessary optical inputs for VLIDORT to return simulated radiances and Jacobians for our retrieval trials. More details on optical property setups for VLIDORT may be found in the review literature (Spurr and Christi, 2019).

**Appendix C. Determination of CSI Threshold**

To identify Hunga aerosol plume pixels and reduce interference with tropospheric clouds, we use a Cloud Screening Index (CSI), which is defined as a TROPOMI radiance ratio at a specific wavelength below 300nm. The CSI wavelength and

 

threshold were determined empirically. Radiance-ratio maps were generated at 0.5 nm intervals between 280 nm and 330 nm and examined (see Supplement S1). We found that short band 1 wavelengths fail to fully capture the Hunga plume, while
longer UV2 wavelengths are affected by tropospheric clouds more than by Hunga aerosols (Figures 2–4). Based on this analysis, radiance ratios at 296, 297, and 298 nm were selected as candidates for the CSI representative wavelength, since they minimized interference with tropospheric clouds while retaining good sensitivity to the Hunga volcanic aerosols. Figure C1 shows retrieved Hunga AOD maps filtered using the CSI at the candidate wavelengths (296, 297, and 298 nm) with CSI thresholds set at 1.05 and 1.1. Areas marked with black circles indicate regions influenced by tropospheric clouds. The
threshold of 1.05 was found to be too low to effectively filter out tropospheric clouds at representative wavelengths. When the threshold was increased to 1.1, filtering at 297 nm and 298 nm still left some tropospheric cloud pixels, whereas filtering at 296 nm screened out most tropospheric clouds and captured most of the Hunga aerosol plume pixels (see Figure 3). Therefore, the radiance ratio at 296 nm was selected as the CSI wavelength, with the associated threshold set to be 1.1.

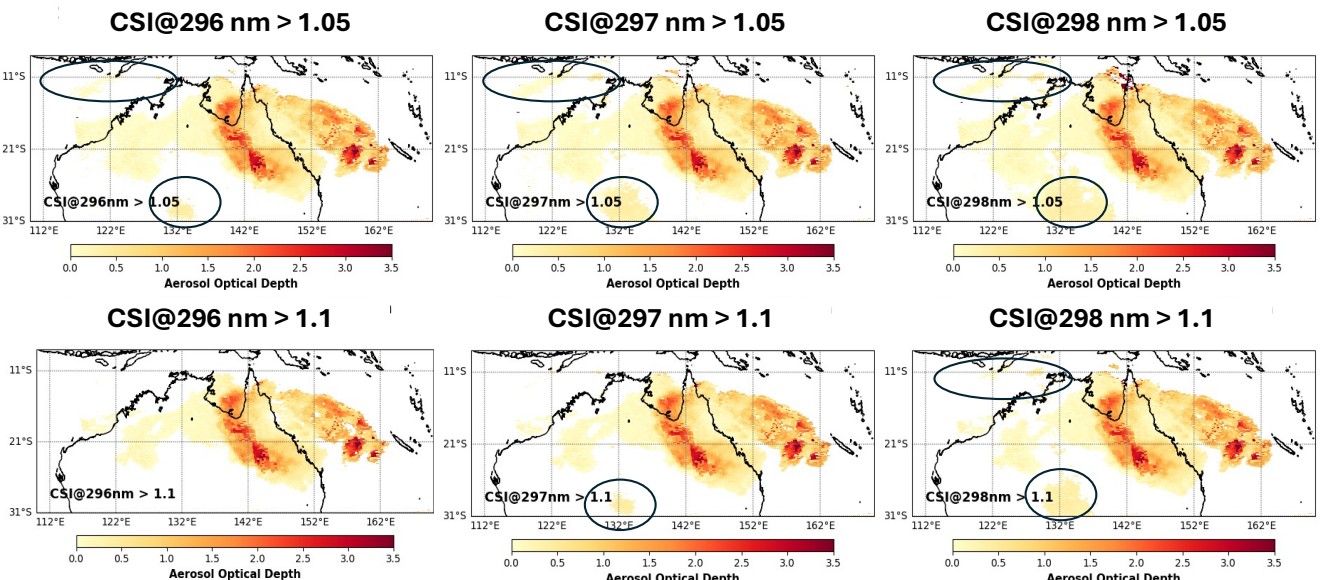

**Figure C1: Hunga-retrieved AOD maps generated by filtering pixels based on radiance-ratio thresholds (1.05 and 1.1) at three candidate CSI wavelengths of 296, 297, and 298 nm. Black circles indicate areas mainly influenced by tropospheric clouds.**


## Appendix D. NASA Goddard Trajectory Calculation of Hunga Aerosol Transport

The "ftraj" trajectory model from NASA's Goddard Space Flight Center Atmospheric Chemistry and Dynamics Laboratory uses a fourth-order Runge–Kutta integration scheme to track parcels isentropically, with optional diabatic adjustments (Schoeberl and Sparling, 1995). The model is driven with winds at 0.25° horizontal resolution and spaced every 6 hours, from





the Goddard Earth Observing System (GEOS) forward-processing system produced by the NASA Global Modeling and Assimilation Office (GMAO).

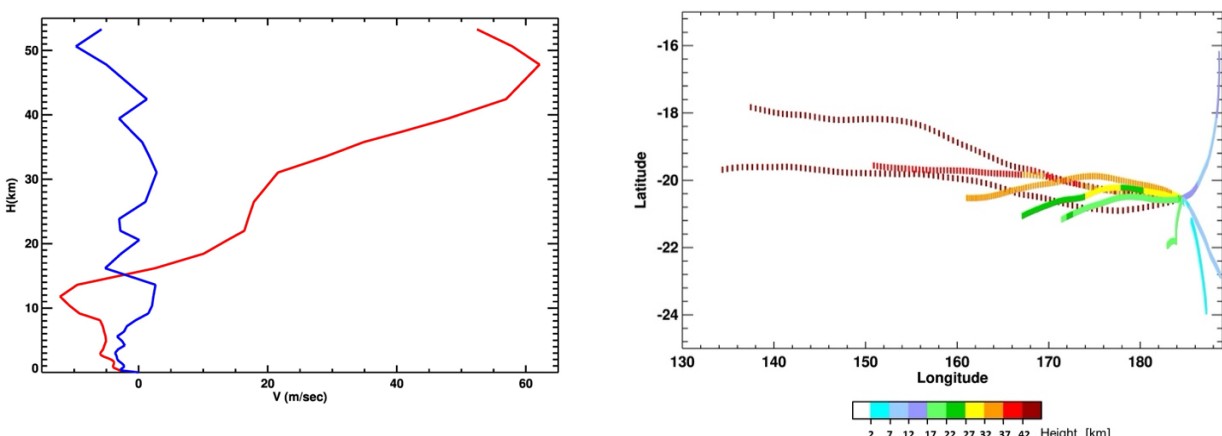

**Figure D1: (Left) Wind vertical profile near the Hunga volcano at the time of the eruption. The blue curve shows the speed of the meridional wind, the red curve the speed of the zonal wind. Clearly the meridional wind is weak at all altitudes, while the easterly winds increase with altitude. (Right) Forward trajectories of air parcels that start from a location directly above the Hunga volcano at different heights, followed through 24 hours using MERRA2 reanalysis winds.**

Figure D1 (right) shows several trajectories calculated using the "ftraj" model. Parcels are started from nadir locations above the eruption point at different heights. In full accordance with the wind field, the tropospheric part of the volcanic plume at altitudes less than 17 km slowly drifts eastward, while the stratospheric part at altitudes higher than 17 km quickly moves westward. At an altitude of ~20 km, the westward drift speed is 10–15 m/sec, reaching 20 m/sec at an altitude of 30 km.

**Appendix E. Hunga Aerosol Retrieval from NOAA-20 OMPS-NP measurements**

The NOAA-20 OMPS Nadir Profiler (OMPS-NP) provides backscattered ultraviolet (BUV) radiance spectra in the nadir viewing direction with a higher signal-to-noise ratio (SNR), but a lower spectral resolution (FWHM~1 nm) than correspondingly for TROPOMI in the band 1 and band 2 UV spectral regions. Given the high SNR and spectral coverage of OMPS-NP, we conducted independent retrievals of AOD and aerosol layer height ($Z_p$) within the Hunga plume. Figure E1 shows (left) the radiance ratio map (CSI at 296 nm) and (right) spectral radiance ratios along the Hunga plume orbit (o21577),

referenced to a background orbit (o21575) on January 17, 2022. The enhancements of radiance ratios are consistent with the TROPOMI radiance ratio patterns presented in Fig. 1. The clear enhancement of OMPS-NP radiance ratios within the Hunga plume and the high SNR of OMPS-NP indicate that OMPS-NP data are sensitive enough to retrieve AOD and $Z_p$. The same





forward model inputs as described in Section 3 (e.g., corrected ozone profiles from M2-SCREAM and aerosol microphysical properties) were used in the OMPS-NP retrievals. Since spectral SNR values are required to construct the measurement and

forward-model error covariance matrix $\boldsymbol{S_\epsilon}$ (as noted in Section 3.3), and considering the noteworthy stray light rejection characteristics of the OMPS-NP instrument, the spectral SNR of OMPS-NP was assumed to be five times higher than the TROPOMI band 1 SNR.

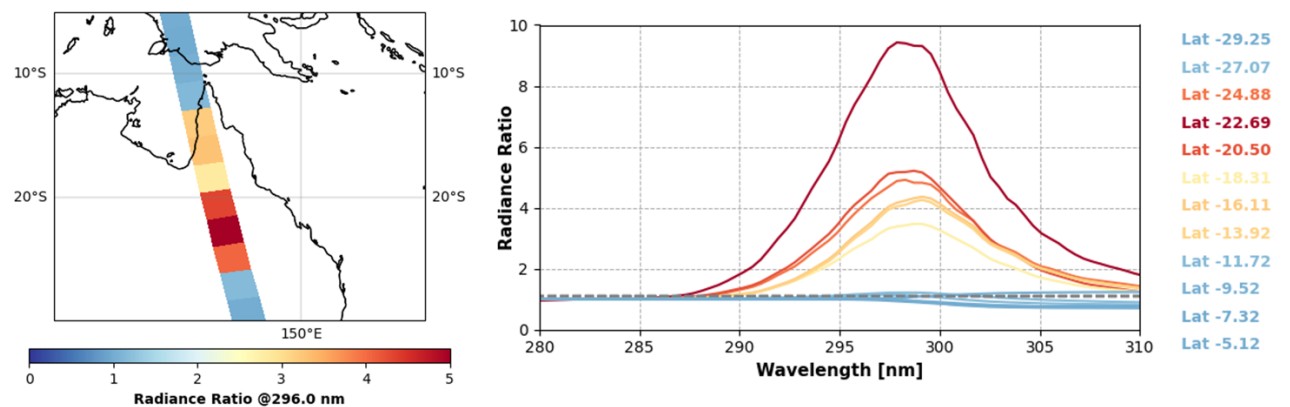

**Figure E1: (Left) OMPS-NP CSI map (radiance ratio at 296 nm) for the plume orbit (o21575) on January 17, 2022. (Right) OMPS-NP spectral radiance ratios (280–310 nm) along the plume orbit (along-track: 35–46; Latitudes: 29.25°S to 5.12°S) on January 17, 2022. Same color scale is applied to both plots. Spectral radiance ratios were derived from the ratio between the plume orbit (o21577) and the background orbit (o21575).**

Figure E2 shows the retrieved AOD and $Z_p$ maps from OMPS-NP and TROPOMI along with their absolute and percentage differences. Both retrievals were performed assuming $n_r$ = 1.47. Smaller TROPOMI pixels were aggregated within OMPS-NP pixels, and to ensure a reasonable comparison, we excluded cases where the number of TROPOMI Hunga pixels with CSI > 1.1 was less than 20% of the total number of collocated pixels. Retrieved AOD and $Z_p$ values are compared in Table E1 for OMPS-NP pixels 37 to 42. As shown in Fig. E2, the spatial distributions of $Z_p$ and AOD from OMPS-NP and aggregated

TROPOMI pixels show good agreement. OMPS-NP $Z_p$ values are slightly lower than those of TROPOMI with an absolute difference (OMPS-NP minus TROPOMI) of 0.25 ± 0.15 km, with an averaged absolute percentage difference of ~0.8%, indicating excellent consistency in $Z_p$ retrievals between the two sensors. However, OMPS-NP AOD values are approximately ~20% higher than those from TROPOMI (0.24 ± 0.17). This result suggests that the higher SNR of OMPS-NP better capture enhanced aerosol signals, while TROPOMI still retrieves $Z_p$ values comparable to OMSP-NP.




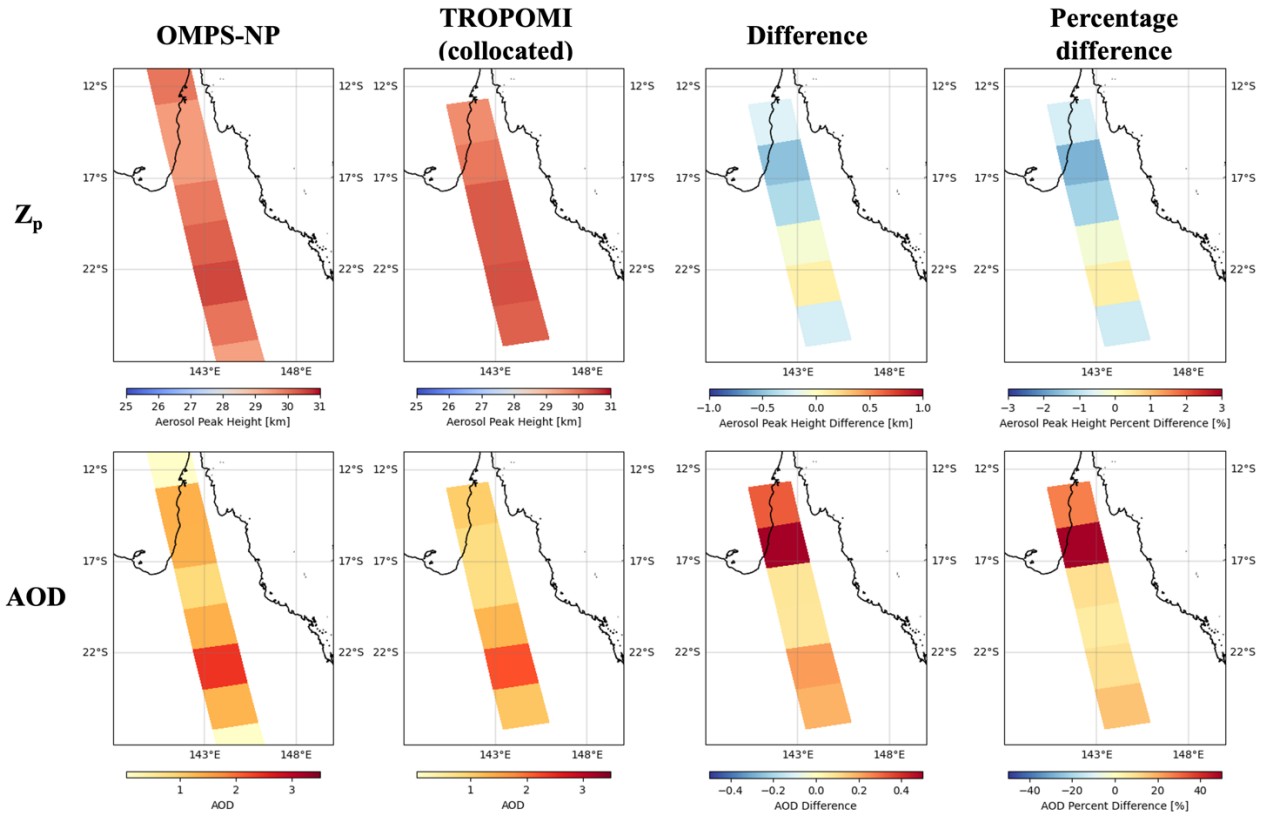

**Figure E2: (Upper row)** Aerosol peak height ($Z_p$) maps retrieved from OMPS-NP and TROPOMI measurements assuming $n_r$ = 1.47, along with absolute and percentage differences between the two retrievals. **(Lower row)** Same as upper panel, but for the AOD. TROPOMI $Z_p$ and AOD values were collocated to match each OMPS-NP along-track location.

**Table E1: Comparison of retrieved Hunga AOD and Aerosol Peak Height ($Z_p$) between NOAA-20 OMPS-NP and TROPOMI, for selected along-track OMPS locations from 37 to 42.**

| | NOAA-20 OMPS-NP | | TROPOMI (collocated) | |
|---|---|---|---|---|
| along-track (0-based) | AOD | $Z_p$ [km] | AOD | $Z_p$ [km] |
| 37 | 1.30 | 30.00 | 1.11 | 30.24 |
| 38 | 2.42 | 30.52 | 2.20 | 30.43 |
| 39 | 1.36 | 30.24 | 1.29 | 30.31 |
| 40 | 0.86 | 29.94 | 0.78 | 30.32 |
| 41 | 1.35 | 29.46 | 0.79 | 29.96 |
| 42 | 1.35 | 29.47 | 1.03 | 29.68 |



**Code availability**

The VLIDORT RT model and the Mie code used in this work are publicly available free of charge, and can be obtained by contacting R. Spurr at RT Solutions, Inc. The retrieval package is governed by the GNU Public License Version 3.0 and will
be placed in a GitHub venue.

**Data availability**

TROPOMI data are publicly available from the Sentinels portal  https://sentinels.copernicus.eu/data-products.
The reprocessed M2-SCREAM output used in this paper is available upon request from Krzysztof Wargan (krzysztof.wargan-1@nasa.gov). NASA ground-based AERONET data are available from https://aeronet.gsfc.nasa.gov/

**Video supplement.** Supplement S1: Spectral Solar Backscattered Ultraviolet (BUV) Radiance Ratios Showing Mid-Stratospheric Aerosols from the January 15, 2022 Hunga Eruption, as Observed by the Copernicus Sentinel 5 Precursor TROPOspheric Monitoring Instrument (TROPOMI) on January 17, 2022. (https://doi.org/10.5446/70186 )

**Author contributions:**

| Contributor role | Role definition | HTHH paper authors |
|---|---|---|
| **Conceptualization** | Ideas; formulation or evolution of overarching research goals and aims. | NAK, OT, SC |
| **Data curation** | Management activities to annotate (produce metadata), scrub data, and maintain research data (including software code, where it is necessary for interpreting the data itself) for initial use and later reuse. | NG, KW |
| **Formal analysis** | Application of statistical, mathematical, computational, or other formal techniques to analyse or synthesize study data. | RS, WC, MC, ESY, OT, SDP, SC, JPV |
| **Funding acquisition** | Acquisition of the financial support for the project leading to this publication. | NAK |
| **Investigation** | Conducting a research and investigation process, specifically performing the experiments, or data/evidence collection. | NAK, WC, MC, NKr, ESY, DH |
| **Methodology** | Development or design of methodology; creation of models. | RS, NK, MC, DL, DH |





| Project administration | Management and coordination responsibility for the research activity planning and execution. | NAK |
|---|---|---|
| Resources | Provision of study materials, reagents, materials, patients, laboratory samples, animals, instrumentation, computing resources, or other analysis tools. | NAK |
| Software | Programming, software development; designing computer programmers; implementation of the computer code and supporting algorithms; testing of existing code components. | RS, MC, NG, ESY |
| Supervision | Oversight and leadership responsibility for the research activity planning and execution, including mentorship external to the core team. | NAK |
| Validation | Verification, whether as a part of the activity or separate, of the overall replication/reproducibility of results/experiments and other research outputs. | NAK, NG, AV, SC |
| Visualization | Preparation, creation, and/or presentation of the published work, specifically visualization/data presentation. | WC, KW, DH |
| Writing – original draft preparation | Creation and/or presentation of the published work, specifically writing the initial draft (including substantive translation). | DH, NAK |
| Writing – review & editing | Preparation, creation, and/or presentation of the published work by those from the original research group, specifically critical review, commentary or revision – including pre- or post-publication stages. | RS, NAK, WC, MC, CL, NKr, AV, KW, SC, JPV, PB |



**Competing interests:**

Some authors are members of the AMT editorial board.

**Acknowledgements**

Support for this work comes through the NASA contract NNG17HP01C. NAK acknowledges additional support from NASA Earth Sciences Division MEaSUREs program. Won-Ei Choi was supported by an appointment to the NASA Postdoctoral
Program at the Goddard Space Flight Center, administered by Oak Ridge Associated Universities under contract with NASA. NAK, WC and OT thank Pete Colarco, Parker Case, Mian Chin, Alexander Smirnov, Thomas Eck, and AERONET team for useful discussions.

The KNMI contributions to this work have been funded by the Netherlands Space Office (NSO), as part of the TROPOMI Science Contract. This publication contains modified Copernicus Sentinel data. Sentinel-5 Precursor is an ESA mission
implemented on behalf of the European Commission. The TROPOMI payload is a joint development by the ESA and the Netherlands Space Office.

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
