# Peer review of "Solar Backscatter Ultraviolet (BUV) Retrievals of Mid-Stratospheric Aerosols from the 2022 Hunga Eruption"

_EGUsphere, 2025_

## Author Comment (AC1)

**Reviewer's comment 1 (RC1)**

This is a very thorough and careful study of the retrieval of aerosol properties in the early stratospheric plume that followed the Hunga eruption on 15 January 2022. It contains a number of technical aspects (this is OK in AMT) which are usually well explained and several appendices in support (that I confess I did not check). I like a lot many points of the discussions. It should be published but I have a number of mostly minor comments and suggestions for improvements that I would like the authors to address before. They are listed in order of appearance and not of importance. All references, those already included and those not, are listed at the end for convenience.

*We sincerely thank the reviewer for the exceptionally detailed and thoughtful review. The reviewer's insightful comments and constructive suggestions have provided valuable guidance that enhanced the clarity and overall quality of the manuscript. We address each point raised in the detailed responses in italic font.*

**RC1-Q1. Figure 1: Please mention the time for each swath and track shown in this figure. It is quite important to account that they are not simultaneous and that the plume was moving quite fast at about 20 angular degrees on the average. This is discussed around Figure 7 but it should be mentioned earlier.**

*Answer: We have added the overpass times for each swath and track in the Figure 1 caption and clarified in the text that the observations are not simultaneous. We also note that the different observation times provide complementary snapshots of the rapidly-evolving plume, which was moving at ≈20° longitude per day (see revised text around Fig. 1) as below:*

*Figure 1 caption: "... The solid line with colored segments shows the suborbital track of the NOAA-20 OMPS-NP (~4:00–4:09 UTC). The inset panel at bottom left shows the variation of spectral radiance ratio with latitude measured by OMPS-NP in the Hunga aerosol plume. Dashed lines are CALIOP ground tracks—blue for daytime (~5:21–5:32 UTC) and black for nighttime (left: ~17:54–18:05 UTC, right: ~16:15–16:26 UTC). The yellow star indicates the location of the Aerosol Robotic Network (AERONET) Lucinda site (18.5198°S; 146.3861°E; elevation: 8.0 m)."*

*Page 3, Lines 90–94 of the revised manuscript: "Also indicated in Fig. 1 are the overpass times of key satellite tracks: the National Oceanic and Atmospheric Administration (NOAA)-20 Ozone Mapping and Profiler Suite (OMPS) Nadir Profiler (NP) swath (~4:00–4:09 UTC), the Cloud-Aerosol Lidar with Orthogonal Polarization (CALIOP) daytime track (~5:21–5:32 UTC), and nighttime tracks (~16:15–16:26 UTC and ~17:54–18:05 UTC). These times indicate that the observations were not simultaneous, but together they provide a complementary view of the rapidly evolving plume (≈20° longitude per day; see also Fig. 7)."*

**RC1-Q2. Longitudes are missing on figure 1. Why is the location of Lucinda indicated as approximate?**

*Answer: Longitudes have been added to Figure 1 as well as C1 and C2 labels. The word "approximate" has been deleted from the caption and the specific coordinates of the Lucinda station (18.5198°S; 146.3861°E) are now provided in the revised Figure 1 caption.*

[Figure]

Figure 1: (a) Shortwave BUV radiances (270–310 nm) of aerosol-rich and background (aerosol-free) regions. (b) Spectral radiance ratios (aerosol/background). (c) True-color NOAA-20 VIIRS map of Australia and the Coral Sea on January 17, 2022. The solid line with colored segments shows the suborbital track of the NOAA-20 OMPS-NP (~4:00–4:09 UTC). The inset panel at bottom left shows the variation of spectral radiance ratio with latitude measured by OMPS-NP in the Hunga aerosol plume. Dashed lines are CALIOP ground tracks—blue for daytime (~5:21–5:32 UTC) and black for nighttime (left: ~17:54–18:05 UTC, right: ~16:15–16:26 UTC). The yellow star indicates the        location of the Aerosol Robotic Network (AERONET) Lucinda site (18.5198°S; 146.3861°E; elevation: 8.0 m).

**RC1-Q3. In general, please consider making fonts for labels larger on all figures. AMT uses a two-column layout and some axes may become hardly readable if reduced in size.**

*Answer: The font sizes for labels and axes have been increased in all figures in order to ensure full readability in the AMT two-column layout.*

**RC1-Q4. This figure shows that there were two components in the plume, which are also seen in Carn et al. (2022) and labelled C1 (western cloud) and C2 (eastern cloud) in Legras et al. (2022), with different altitude and composition and which evolved in a different way over the first weeks after the eruption until they mixed. Unfortunately, this difference is not clearly conveyed in the analysis and the discussion. The C1 cloud is aerosol rich because it is also water rich and the conversion from SO2 to sulfate occurred faster than in C2. It also descended faster because of stronger water radiative cooling.**

*Answer: We have added C1 and C2 labels in Figure 1. We revised the text in the Introduction and Results to clarify the distinction between the two plume components (C1 and C2) as follows:*

*Introduction (Section 1) - Page 3, Lines 85–90: "The approximate locations of two distinct Hunga plumes are outlined over the Queensland region of Australia (Aerosol-rich plume; C1) and over the Coral Sea ($SO_2$–rich plume; C2) – see Fig. 5d in Carn et al. (2022) and Figs. 4b–d in Legras et al. (2022). These two plumes showed different altitudes and compositions and evolved differently during the first weeks after the eruption before eventually mixing. The aerosol-rich C1 plume was also water-rich, which likely accelerated the $SO_2$-to-sulfate conversion and enhanced radiative cooling, thus contributing to its more rapid descent compared to that of the $SO_2$-rich C2 plume."*

*Result (Section 4.1) – Pages 17-18, Lines 487–493: "This west–east contrast is consistent with previous studies distinguishing the western aerosol- and water-rich cloud (C1) and the eastern $SO_2$-rich cloud (C2) (Carn et al., 2022; Legras et al., 2022). The C1 cloud likely experienced faster $SO_2$-to-sulfate conversion and stronger water-driven radiative cooling, which contributed to its more rapid descent compared to that for C2, as noted in previous analyses (Legras et al., 2022; Sellitto et al., 2022)."*

**RC1-Q5. It is important to mention in section 2 or 3 that the injected water vapour had strong local effects on the temperature of the stratosphere that were not represented correctly by standard assimilation systems which did not assimilate water vapour data. After a while the assimilation of temperature data recovers correct profiles even if the budget was wrong (cf Coy et al., 2022) but large errors occurred during the first days. This is also because the GPS radio-occultation signal was strongly contaminated by the unusual high level of water in the stratosphere (Randel et al., 2023). Some data were rejected but not all. Since M2-SCREAM assimilates water from MLS, it has largely corrected these effects and exhibits an equilibrated heat budget (Coy et al., 2022). However, I would not be confident that everything was perfectly right over the first few days after the eruption and I recommend to check the temperature profiles from high-quality radiosoundings available from Australian stations (Vömel et al., 2023). During that period the two plumes descended very fast, especially C1, owing to the cooling by water vapour (Sellitto et al., 2022).**

*Answer: The authors agree that the injected $H_2O$ could have produced strong local, short-term temperature perturbations that were not perfectly represented by standard assimilation systems without stratospheric $H_2O$ assimilation. In Section 2, we have now remarked that in the first few days after the eruption, the temperature field may carry higher uncertainty because (i) assimilated $H_2O$ observations were sparse (with gaps) and (ii) GPS radio-occultation retrievals were partially contaminated by the unusually high stratospheric moisture, with some data rejected but not all (Randel et al., 2023). In contrast, M2-SCREAM assimilates MLS $H_2O$ and recovers a balanced heat budget on longer time scales (Coy et al., 2022), consistent with the literature. At the same time, we do not see a clear temperature anomaly on January 17, 2022, in the reanalysis. Given the large volume of assimilated microwave and hyperspectral radiances, the analyzed temperatures are expected to be well constrained, and any early bias is likely small on the two-day time scale in our study (radiative damping times are typically on the order of 10–20 days). Therefore, we expect minimal impact on our retrievals and conclusions.*

*Based on the reviewer's comment, the authors have added the following clarification in Section 2.2:*

*Page 10, lines 243–250: "Regarding M2-SCREAM temperature profiles, early-January temperatures may be less reliable due to sparsely assimilated $H_2O$ and partial GPS radio-occultation contamination under conditions of extreme moisture (Randel et al., 2023).To assess the reliability of these temperatures, we validated the M2-SCREAM profiles using radiosonde data from stations in Australia and New Caledonia (University of Wyoming archive;* https://weather.uwyo.edu/upperair/sounding.shtml*). We selected eight cases in which the relative humidity exceeded 10% at least once above 15 km, including the January 17 2022 00 UTC sounding. The M2-SCREAM temperatures agreed with the radiosonde measurements within ~4 K over 15−30 km range, supporting that the re-analyzed M2-SCREAM temperature profiles are sufficiently reliable for interpreting the early Hunga plume conditions."*

[Figure]

**RC1-Q6. L272-275: These details can be left for the appendix. Once Jacobians have been mentioned, it is clear that linearization is required.**

*Answer: We have clarified text in this section, and some of the text in this paragraph was moved to Appendix B. More specifically, the following paragraph has been moved to Appendix B:*

*Page 30, lines 819–822: "For radiance simulations, it is necessary to construct an input set of total optical properties (optical thickness values, single-scattering albedos, spherical-function expansion coefficients, scattering matrices) for VLIDORT. In addition, for calculations of associated Jacobians with respect to aerosol retrieval parameters, VLIDORT requires an additional set of linearized total optical property inputs. Determination of VLIDORT optical property inputs is discussed in this Appendix."*

**RC1-Q7. Section 3.3: The iterative procedure is described but the first guess is also important as the minimization path can lead to spurious minima. It is not indicated how this problem is avoided. Perhaps it does not occur in this application but often look-up tables are needed to provide a good enough first guess.**

*Answer: First, the Levenberg-Marquardt (L-M) Inverse process effectively starts out as a Steepest Descent Method (which converges more slowly, but globally) and then, through the use of the Levenberg-Marquardt parameter, switches over to a Quasi-Newton Method (which converges more quickly, but locally). The use of the L-M process thus helps guard against convergence to spurious minima in parameter space. To check the robustness of the L-M inversions, we did do some simulations in which we varied the first-guess values of the state vector elements, in order to check that the retrievals still converged to the same final state vector solution. For the cases ran, it was found the retrievals were stable in this regard. These results are below:*

| Some "Initial Guess" Tests | | | | |
|---|---|---|---|---|
| *TROPOMI Orbit / Test* | *Initial AOD* | *Initial Peak Height* | *Final AOD* | *Final Peak Height* |
| Orbit 22086 Case | | | | |
| Test 1 | 2.0 | 29.0 | 1.90209 | 30.24126 |
| Test 2 | 3.0 | 30.0 | 1.90212 | 30.24124 |
| Test 3 | 4.0 | 31.0 | 1.90223 | 30.24113 |
| Orbit 22087 Case | | | | |
| Test 1 | 2.0 | 29.0 | 1.93108 | 30.21280 |
| Test 2 | 3.0 | 30.0 | 1.93112 | 30.21277 |
| Test 3 | 4.0 | 31.0 | 1.93140 | 30.21254 |

**RC1-Q8. L330: I would say the cloud is more centered at 30 km than at 32 km. A single daily crossing of C1 is here used but there are other relevant CALIOP data for the same day that should be used, especially because they are on night orbits with much better S/N. More precisely, there is a crossing of C2 at 16:19 UTC (orbit 2022-01-17T15-45-52ZN) and a crossing of C1 at 17:57 UTC (orbit 2022-01-17T17-24-22ZN). Of course, the signal is strong enough to see the plume clearly on daily data but night data allow to check the low value of**

**depolarization which supports spherical liquid sulfates and the absence of solid particles. The large signal and the isolation of the plume layer leads also to a determination of the AOD (Duchamp et al., 2025) that could be used in section 4.2.**

*Answers: 1) The authors corrected the typo in peak altitude to 30 km - see Figure 6c in the revised manuscript. We have also added two nighttime CALIOP crossings in Fig. 6d and 6e as suggested.*

*2) We compared depolarization ratios (DR) for nighttime crossings (i.e., Figs 2c and 2d in Duchamp et al., 2025) with the daytime crossing (left panel). We note that daytime CALIOP DR data are noisier but have the advantage of the minimal time difference with our TROPOMI retrievals. We did not include these DR figures in the text but added the reference in section 4.2.*

[Figure]

*Figure. The daytime (left) and nighttime CALIOP DPs curtains show low depolarization, which supports spherical particles in both C2 (middle) and C1 (left and right) Hunga clouds. However, the daytime C1 data (right) show somewhat higher DR with larger noise as well as some low parts of the C2 cloud – compare fig 2c and 2d in Duchamp et al., 2025.*

*We thank the reviewer for suggestion to compare with the nighttime CALIOP AOD retrievals (Duchamp et al., 2025). We note that the timing mismatch with the TROPOMI overpass, CALIOP's very narrow swath, and the pronounced spatial heterogeneity within the Hunga plume make a pixel-wise AOD comparison challenging. We added the following text in section 4.2:*

*Page 21, Lines 610–614: "Recent AOD retrievals from CALIOP nighttime Hunga overpasses estimate somewhat larger $AOD_{532}$ values of ~ 1.24±0.13 (1σ) for C1 and ~ 1.01±0.12 (1σ) for C2 on January 17 (Duchamp et al., 2025). These CALIOP-derived $AOD_{532}$ values are still qualitatively consistent with TROPOMI results, considering the spatial heterogeneity, the temporal separation between the*

*overpasses, and the spectral differences between the retrievals. A more comprehensive comparison with CALIOP will be conducted in a follow-up study".*

**RC1-Q9. L.341. There is a known thermodynamic dependency of wt on temperature and humidity for which a suitable parameterization is available in Tabazadeh et al. (1997). This might significantly restrain the range of values to be considered and perhaps give a way to a more reliable estimate than what is done in section 4.4.**

*Answer: In order to determine physically plausible lower and upper bounds for wt%, the authors have added Appendix F, which contains results from simulations based on the GEOS Chemistry–Climate Model (GEOS-CCM) coupled to CARMA (Case et al., 2023). In this configuration, $H_2SO_4$ wt% is calculated from temperature and relative humidity using the parameterization of Tabazadeh et al. (1997). On January 17, 2022, the model shows zonal-maximum RH ≈ 60% near ~20°S, indicating strong moistening in the plume core and yielding $H_2SO_4$ wt% as low as ~40% in the core (Fig. F1). Above ~32 km, the simulated $H_2SO_4$ wt% increases to ≥ 80%. These results independently support the lower and upper bounds used in our retrievals, i.e. 30 and 80 wt% respectively; the results also justify our choice of a mid-range representative value ~50wt% in the analysis.*

[Figure]

*Figure F1: (Left) Zonal maximum relative humidity on January 17, 2022 in NASA GEOS. (Right) Sulfate weight percentage of total aerosol mass (wt%) associated with zonal maximum relative humidity.*

*Case, P., Colarco, P.R., Toon, B., Aquila, V. and Keller, C.A., 2023. Interactive stratospheric aerosol microphysics-chemistry simulations of the 1991 Pinatubo volcanic aerosols with newly coupled sectional aerosol and stratosphere-troposphere chemistry modules in the NASA GEOS Chemistry-Climate Model (CCM). Journal of Advances in Modeling Earth Systems, 15(8), p.e2022MS003147.*

**RC1-Q10. L348: Sg = 1.545 μm does not make sense for a standard deviation (especially with a median at 0.14 μm) but 1.545 (without dimension) makes perfectly sense as the width parameter of the lognormal distribution. I made a check myself using the Lucinda v2.0 data for 17 January 2022 and I found a better fit to the volume size distribution with a median at 0.13 μm and a width parameter 1.57 but the retained values are probably OK. By the way, Boichu et al. (2023) give 0.22-0.23 μm as the effective radius (I find 0.21 μm) but I do not see where the width 1.545 is mentioned in this paper. They use v1.5 version of the data.**

*Answer: The authors thank the reviewer for pointing out the use of a distance unit for the width parameter ($S_g$); this quantity is now corrected to be dimensionless. The number size distribution parameters ($R_g$ and $S_g$) presented in our paper were calculated from the volume size distribution retrieved at the Lucinda AERONET site using v1.5 inversions and neglecting small tropospheric coarse mode. We have also clarified in the text that our values were obtained using AERONET version 1.5 inversions, consistent with Boichu et al. (2023). To double-check, we fitted the Lucinda data from the quality assured v2.0 for 17 January 2022 inversions and obtained $R_g$ = 0.12 µm, $S_g$ = 1.60, and $r_{eff}$ = 0.21 µm. These values are very close to those reported by the reviewer from v2.0 ($R_g$ = 0.13 µm, $S_g$ = 1.57, $r_{eff}$ = 0.21 µm), and within the expected uncertainties. We conclude that the adopted values of these aerosol parameters are reasonable and that they do not affect the overall retrieval results or the conclusions drawn.*

**RC1-Q11. L355 A more up to date source for high-resolution ozone cross-sections with temperature dependency is Serdyunchenko et al. (2011).**

*Answer: We appreciate the suggestion to use the temperature-dependent high-resolution ozone cross-sections of Serdyunchenko et al. (2011). In our forward model optical property set-up, the ozone cross-sections must be parameterized as a function of temperature. We therefore carried out a quadratic parameterization of the Serdyunchenko dataset (11 temperatures, 193–293 K) based on the equation below:*

$$\alpha(\lambda) = 10^{-20} \cdot [C_0(\lambda) + C_1(\lambda) \cdot T + C_2(\lambda) \cdot T^2]$$

*This is similar to the original parameterization with the older $O_3$ cross-section data set (Brion et al., 1993). Relative to the original Serdyunchenko et al. (2011) values, the parameterization departs by up to ~3.5% below 300 nm, with the largest differences over 289–296 nm; between 300–310 nm the mean difference is ≤ 1%. When we re-ran the retrievals using the parameterized Serdyunchenko cross-sections, the resulting changes were modest: in the AOD −4% (mean) and for peak height $Z_p$ −0.4% (mean); these values resulted in a plume total aerosol mass difference of ~0.02 Tg. Given the strong ozone absorption in our fitting window and the additional uncertainty introduced by the parameterization step, and because the impact on aerosol mass is minor, we have elected to retain the Brion et al. (1993) cross-sections for stability and continuity.*

**RC1-Q12. L395 In the range of altitudes and latitude of the plume the horizontal shear was more significant than the vertical shear. This is visible in the fast deformation of the clouds between images collected on 16 January (e.g. Carn et al., 2022) and the present study.**

*Answer: Following the reviewer's suggestion, the authors have revised the text to explicitly emphasize the role of horizontal shear, as follows:*

*Page 18, Lines 495–497: "The wind profile is such that the stratospheric part of the volcanic cloud moves westwards, while tropospheric parts of the cloud move eastward; this reflects strong horizontal wind shear that was more significant than vertical shear at this latitude and over this altitude range (Carn et al., 2022; Sadeghi et al., 2025)."*

**RC1-Q13. L.404 It is a bit difficult to appreciate the altitude by eye since the colour scale is very smooth in this range of altitudes. Is this estimate of 31 km produced by a spatial average of the data or is it a visual estimate? An other estimate should be given for the C2 cloud and please also exploit the day orbit for C1 in the same way. The clouds are not uniform and the sampling of CALIOP is on a very narrow curtain.**

*Answer: We have adjusted the color scale to improve readability of the peak-height map in the revised version of Figure 7a. We have also clarified that the ~30 km estimate is a TROPOMI-based spatial mean over the C1 area, and we have now provided the corresponding C2 mean using the same procedure.*

*The following text is apposite:*

*Page 18, Lines 497–501: "The higher the altitude, the stronger the easterly winds, so those parts of the C1 cloud at $Z_p$ ~30 ± 1 km were advected westward more quickly than the lower parts (Sadeghi et al., 2025). By contrast, the C2 cloud, centered near $Z_p$ ~25 ± 1 km in the eastern sector, showed slower westward advection consistent with a weaker easterly wind field at these lower stratospheric levels. CALIOP profiles (Fig. 7b and c) confirm the TROPOMI-retrieved Hunga plume heights."*

*Please see the revised Figure 7 below.*

[Figure]

*Figure 7: (a) Retrieved aerosol plume peak height $Z_p$ [km] assuming upper limit refractive index $n_r$ = 1.47, from TROPOMI orbits 22086 (~3:20 UTC) and 22087 (~5 UTC) on January 17, 2022. The dashed magenta line shows a back-trajectory matchup between nighttime CALIOP measurements and daytime TROPOMI Hunga aerosol retrievals. (b) the CALIOP attenuated backscatter during daytime (~5:23–5:34 UTC) with the ground track shown in (a) with the solid red line. (c) same as (b) but for a nighttime track (~16:14–16:25 UTC), which is shown with the solid magenta line in (a). (d) Absolute difference in retrieved $Z_p$ assuming low ($n_r$ =1.39) and high ($n_r$ =1.47) refractive index scenarios.*

**RC1-Q14. L405 Of course, 1 km over 30 km is 3% but what is the range of retrievable altitudes for TROPOMI and the width of the kernels which would provide a better idea of the accuracy.**

*Answer: Averaging kernels are usually studied in the context of performing maximum aposteriori (MAP) retrievals of (1) a profile of atmospheric constituents where (2) apriori statistics are in use [see e.g. Rodgers' Inverse Methods for Atmospheric Sounding (2000) pages 37, 56 and 67]. Related to this, apriori information using a covariance matrix is most appropriately applied in cases where measurements have been made over time and accompanying statistics compiled. In the case of the Hunga Tonga (HT) eruption, we are talking about an extreme event that is two to three standard deviations from the mean state of the atmosphere in that area of the world. Thus, even if general aerosol statistics exist, such apriori would be inappropriate since they would serve to over-constrain the aerosol retrievals we are seeking to perform, given the exceptional nature of the HT event.*

*Thus, we opted for doing a more column-oriented maximum likelihood (ML) retrieval where apriori covariance matrices are not used. However, even here we do not operate without applying some form of apriori constraint; here, it is in the form of the Gaussian-like shape of the HT aerosol plume in the vertical. This is a compromise position: it has the drawback of not allowing as many degrees of freedom in the vertical during the retrieval, but the benefit of reducing instability in the vertical due to measurement noise that might otherwise occur since we are operating without an apriori covariance.*

*However, even though not used in our formal, more column-oriented retrievals, more layer-oriented studies have been performed to get some sense of the vertical range and resolution of the observing system. Given that we do not have the smoothing influence of an apriori covariance matrix operating in those studies, the vertical resolution is more closely, though not entirely, tied to the vertical resolution of the atmospheric grid we are using (in this case, 0.25km / atmospheric layer). Based on those studies, the current system appears to have an estimated effective range of approximately 25km to 43km in the vertical with an effective resolution of 0.3-0.4km.*

**RC1-Q15. Section 4.4: ). Carn et al. (2022) estimated a $SO_2$ e-folding time of $\tau$=6 days based on satellite estimates of the $SO_2$ column and used this to estimate a total emitted mass of about 0.45 Tg of $SO_2$ from the 15 January eruption. However, this analysis did not separate the C1 and C2 clouds, and the lifetime of $SO_2$ in C1 was likely shorter, which suggests an underestimation of the total injected mass reported by Carn et al. (2022). Even if a new estimate is not done here, this issue might be considered.**

*Answer: We agree that applying a single e-folding time ($\tau$=6 days) to the combined C1+C2 clouds could bias the initial $SO_2$ mass at the low end, if C1 decayed faster. In Section 4.4, we constrain the January 17 sulfate aerosol composition by matching the expected sulfur (S) aerosol mass (computed from half of average satellite $SO_2$ retrievals and $\tau$ = 6 reported in Carn et al., 2022) with retrievals performed at the limiting sulfuric acid weight percent wt% bounds (30% and 80%); this points to a representative value of wt%~50%.*

*To check this independently, we have added Appendix F, which describes a GEOS-CCM–CARMA Hunga aerosol simulation initialized with 0.5 Tg $SO_2$ and ≈150 Tg $H_2O$ (consistent with MLS). On 17 January*

*2022, the model diagnosed wt% from temperature and relative humidity (Tabazadeh et al., 1997) and yielded 40–80% across the plume, with ~50% in the core. This model-based composition agrees with our mass-constrained inference.*

**RC1-Q16. L535: I give little trust to this estimate of wt by such indirect way which is subject to large uncertainties (see above). I would prefer an estimate based on thermodynamic data as mentioned above.**

*Answer: We appreciate the concern. As noted in RC1-Q9 and Q15, in the revised manuscript we now base the wt% estimate on thermodynamic data (Tabazadeh et al., 1997; Case et al., 2023), as recommended.*

**RC1-Q17. Bruckert et al. (2025) is now published. Do not forget to update the reference.**

*Answer: Thank you for the update. We have updated the reference in the revised manuscript.*

---

## Author Comment (AC2)

**Reviewer's comment 2 (RC2)**

This is an excellent and comprehensive paper that shows that stratospheric aerosol parameters can be retrieved from nadir backscatter measurements for a large volcanic plume that is above the ozone layer. The paper is well written, and the problem is approached systematically with detailed explanations and appropriate appendices. The paper is highly appropriate for the scope of AMT and needs very little revision for publication. A few minor comments are included below for the authors to address in a revision.

*We sincerely thank the reviewer for the thoughtful review. We address each point raised in the detailed responses below.*

**RC2-Q1. The definition of wet aerosol mass (in the abstract) should be clarified (and why is "wet" in quotations?)**

*Answer: We agree and removed quotation marks in the revised manuscript. We have clarified that wet aerosol mass refers to the aerosol column mass of aqueous sulfate droplets, i.e., including water uptake at stratospheric temperatures and relative humidities. The altered text is as follows:*

*Abstract-Page 2, Lines 45–46: "We estimate the total Hunga stratospheric wet aerosol mass (sulfate solution droplets, including water uptake) to be $M_{aer} \sim 0.5 \pm 0.05$ Tg."*

*Page 24, Lines 659–660: "~ a corresponding wet aerosol mass (~0.47 Tg), where "wet" denotes aqueous sulfate droplets including water uptake at stratospheric relative humidities."*

**RC2-Q2. Section 3.1 Normalization to background radiances includes a correction for ozone profile differences but it is highly feasible that air density differences are not characterized by M2-SCREAM in the early plume. What is the potential impact?**

*Answer: We agree that M2-SCREAM may not fully represent plume-related density changes in the first days after the eruption. In principle, this could affect Rayleigh scattering (Bodhaine et al., 1999) and the scaling of ozone absorption (Guo and Lu, 2006), since both depend on air density. These effects could add some uncertainty to the normalized radiances. To this end, we have added the following paragraphs to the paper, and added the reference to (Guo and Lu, 2006).*

*However, during the course of this research, we performed a study to investigate the influence of variations in atmospheric density profile and ozone profile for different scenes characteristic of the period following the eruption. For a given set of TROPOMI BUV radiances, the following tests were done:*

- *the original atmospheric temperature and pressure profiles of the accompanying scene were replaced while holding the ozone profile fixed;*
- *the original atmospheric ozone profile of the accompanying scene was replaced while holding the temperature and pressure profiles fixed;*
- *the original set of all three were replaced to observe their combined influence.*

*For each of these combinations, several retrievals were done to observe the influence of these variations on the retrieval results. Replacement of the ozone profile was found to generate the greatest influence, and therefore we paid much more attention to defining this profile for each TROPOMI scene. However, differences in atmospheric density profiles due to changes in temperature and pressure were found to generate small effects (on the order of a few tenths in retrieved AOD and sub-kilometer changes in peak height).*

*The authors added the following sentences in the revised manuscript:*

*Page 12, lines 275–278: "We also examined the potential impact of possible air-density variations that may not be fully captured by the M2-SCREAM reanalysis during the early days after the eruption. Such differences could influence Rayleigh scattering (Bodhaine et al., 1999) and the scaling of ozone absorption (Guo and Lu, 2006), since both depend on air density."*

*Bodhaine, B. A., Wood, N. B., Dutton, E. G., & Slusser, J. R. (1999). On Rayleigh optical depth calculations. Journal of Atmospheric and Oceanic Technology, 16(11), 1854-1861.*

*Guo, X., and Lu, D.: Feasibility study for joint retrieval of air density and ozone in the stratosphere and mesosphere with the limb-scan technique. Applied optics, 45(35), 9021-9030, 2006.*

**RC2-Q3. Section 3.4 Description of the unimodal particle size distribution parameters does not make sense. What is the sensitivity to the choice of 8 discrete ordinates with delta M scaling?**

*Answer: Regarding the PSD used for the volcanic aerosol, we have corrected the description of the unimodal lognormal size distribution parameters (with the width/variance parameter now properly given as dimensionless).*

*Regarding the number of discrete ordinate streams used in the RT computations, we remark that it is not 8 discrete ordinates in total that is being used, but rather "8 ordinates in the polar half-space" (i.e. 16 streams in total between local nadir to zenith). We have changed the text to clarify this distinction. However, even with the 16 total streams and the delta-M scaling ansatz that were actually used in this work, a comparison was made against using 32 total streams with delta-M scaling (often considered a kind of "gold standard" in these types of retrieval settings) and these yielded negligible differences in spectra.*

**RC2-Q4. Section 3.5 No results or analyses are shown from the validation with synthetic data. Typically a retrieval result and comparison with the input state would be shown just to verify the algorithm fidelity.**

*Answer: To help validate the Hunga-Tonga (HT) retrieval algorithm forward model (FM), we did some initial testing by comparing simulations using our FM with calculations made by the OMI simulator model, for a case with typical atmospheric and surface conditions experienced following the HT eruption during the period of this study (that is, profiles of temperature, pressure and ozone, vertical characterization of the volcanic aerosol, plus aerosol PSD and surface albedo). The resulting spectra were found to be in good agreement. It should be noted that this test actually helped clarify some HT-FM input issues that were addressed at an earlier phase in the study.*

*We also performed some closed-loop tests on the HT retrieval algorithm itself, again for typical post-eruption scenarios encountered in this study. Here, two types of tests were done to test retrieval fidelity, each test based on synthetic spectra generated by the forward model using a pre-defined state vector $X_{true} = \{AOD, Pk\ Hgt\}$.*

- *The first retrieval test used the HT-FM-generated synthetic spectra with no noise; this is to check that the algorithm does indeed retrieve the known state (i.e. $X_{ret} \approx X_{true}$). This test was successful.*
- *The second retrieval test again used the synthetic spectra, but this time including noise levels representative of TROPOMI UV Band 1 measurements; this is to check that the retrieval algorithm can again basically retrieve the known state (i.e. $X_{ret} \approx X_{true}$), but this time observing increases in estimated uncertainty and/or bias. This test was also informative and successful.*

*The authors added more details of synthetic test in Section 3.5 as below:*

*Page 17, lines 466–474: "To further verify the fidelity of the retrieval algorithm itself, we performed a set of closed-loop validation tests using synthetic radiance spectra generated by the forward model. Each test used a pre-defined state vector $X_{true} = \{AOD, Z_p\}$ to represent typical post-eruption conditions. Two cases were examined: (1) a retrieval based on noise-free synthetic spectra to ensure that the algorithm could reproduce the known state ($X_{ret} \approx X_{true}$), and (2) a retrieval using spectra with added random noise levels representative of TROPOMI UV Band 1 measurements to assess retrieval robustness under realistic conditions. In both cases, the retrieved parameters converged closely to the true inputs, demonstrating that the Hunga retrieval framework is stable and internally consistent. The noise-added tests also showed slightly larger retrieval uncertainties, as expected, but without systematic bias in AOD or $Z_p$. Overall, the results increased confidence in the forward and inverse model configurations used in the actual Hunga plume retrievals."*

*In this regard, comments have been added to the end of both sections 3.2 and 3.3 to mention that these kinds of integrity tests were performed. However, we feel that, given the number of figures already present in the paper dealing with uncertainty and/or validation, it was not necessary to present an additional figure or figures specifically related to this question of retrieval integrity.*

**RC2-Q5. Section 4.2 The retrieved AOD varies greatly with the refractive index assumptions. Is this wide range of refractive indices realistic for Hunga?**

*Answer: We acknowledge that variations in the real part of the refractive index between the extreme values n=1.39 (~30wt%) and n=1.47 (~80wt%) have a large effect on the retrieved AOD values, i.e., Figure 8 (bottom).*

*As to whether this wide refractive-index range is realistic for Hunga, we have carried out some further analysis of this issue and added new material. In the revised paper, we have added a new Appendix F, in which GEOS-CCM–CARMA diagnostics (Case et al., 2023) show $H_2SO_4$ wt% ~40–70% on 17 Jan (core ~40%). This composition span is consistent with laboratory refractive-index measurements for sulfuric-acid solutions near 312 nm (e.g., Beyer et al., 1996), and it justifies bracketing the real part of the refractive index to the range 1.39–1.47 for our sensitivity tests.*

*As shown in Section 4.4, we estimate the ambient Hunga aerosol solution concentration to be ~50 wt%. Based on the measurements reported by Beyer et al. (1996), this corresponds to a refractive index of ~1.42 at 312 nm, which is consistent with a solution concentration of approximately 52 wt%.*

*To help the reader to understand our assumptions regarding the wide range of the real part of the refractive index, the following sentence has been inserted in the revised manuscript:*

*Pages 15–16, lines 402–425: "This wide range for the real part of the refractive index reflects the plume composition simulated by NASA Goddard Earth Observing System (GEOS) Earth system model (see Appendix F), i.e., $H_2SO_4$ wt% spanning roughly 40–70% (core values ~40%) and thus reasonably bracketing the ~30–80% range."*

*Beyer, K. D., Ravishankara, A. R., & Lovejoy, E. R. (1996). Measurements of UV refractive indices and densities of H2SO4/H2O and H2SO4/HNO3/H2O solutions. Journal of Geophysical Research: Atmospheres, 101(D9), 14519-14524.*